

# Efficient mutual magic and magic capacity
# with matrix product states

**Poetri Sonya Tarabunga[1,2⋆] and Tobias Haug[3†]**

**1** Technical University of Munich, TUM School of Natural Sciences,
Physics Department, 85748 Garching, Germany
**2** Munich Center for Quantum Science and Technology (MCQST),
Schellingstr. 4, 80799 München, Germany
**3** Quantum Research Center, Technology Innovation Institute, Abu Dhabi, UAE

⋆ poetri.tarabunga@tum.de , † tobias.haug@u.nus.edu

## Abstract

Stabilizer Rényi entropies (SREs) probe the non-stabilizerness (or "magic") of many-body systems and quantum computers. Here, we introduce the mutual von-Neumann SRE and magic capacity, which can be efficiently computed in time $O(N\chi^3)$ for matrix product states (MPSs) of bond dimension $\chi$. We find that mutual SRE characterizes the critical point of ground states of the transverse-field Ising model, independently of the chosen local basis. Then, we relate the magic capacity to the anti-flatness of the Pauli spectrum, which quantifies the complexity of computing SREs. The magic capacity characterizes transitions in the ground state of the Heisenberg and Ising model, randomness of Clifford+T circuits, and distinguishes typical and atypical states. Finally, we make progress on numerical techniques: we design two improved Monte-Carlo algorithms to compute the mutual 2-SRE, overcoming limitations of previous approaches based on local update. We also give improved statevector simulation methods for Bell sampling and SREs with $O(8^{N/2})$ time and $O(2^N)$ memory, which we demonstrate for 24 qubits. Our work uncovers improved approaches to study the complexity of quantum many-body systems.

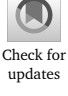

# 1 Introduction

Nonstabilizerness, also known as magic, is recognized as a key resource for quantum computing and a necessary condition for demonstrating quantum advantage [1,2]. It is based on the notion of stabilizer states [3], which are those generated by Clifford unitary operators and which play a key role in quantum computation and error correction [4–6].

Essentially, nonstabilizerness quantifies the degree to which a quantum state cannot be approximated by a stabilizer state. Given that stabilizer states are efficiently simulable classically [7–9], nonstabilizerness determines the minimal non-Clifford resources required for universal quantum computation [5,6]. It also provides a lower bound on these resources and is indicative of the complexity of classical simulation algorithms using techniques based on the stabilizer formalism [2,10]. Consequently, the quantitative characterization of nonstabilizerness, especially in many-body contexts, is of paramount importance.

Several measures of magic have been proposed in quantum information theory, such as the min-relative entropy of magic [11,12], the relative entropy of magic [2], the robustness of magic [12,13], and the basis-minimized stabilizerness asymmetry [14]. However, the computation of these measures involve complex minimization procedures, making their computation a significant challenge in many-body systems. To address this problem, stabilizer Rényi entropies (SREs) [15] were introduced, providing efficiently accessible magic measures for both numerical [16–23] and experimental studies [24–28]. SREs have found importance in many-body phenomena [12], such as phase transitions [16, 19, 21, 23, 29–32], quantum chaos [33–37], quantum dynamics [38–41], transition in monitored circuits [42–44], and the structure of entanglement [45–50].

In addition to the study of magic in the full state, recently there has been a growing interest in the characterization of more refined aspects of magic. In particular, there has been substantial interest in nonlocal magic, which represents the nonstabilizerness that resides within the correlations between the subsystems [19, 44, 51–58]. Although nonlocal magic is ideally defined using a mixed-state magic monotone, the mutual 2-SRE [19] has been hypothesized to capture some form of nonlocal magic. While the SRE is not a proper magic monotone for mixed states, it can be used to construct mixed state magic witness [28]. Notably, in some cases the mutual 2-SRE was found to display the same behavior as the mutual magic defined using genuine mixed state magic monotones [44]. It is also appealing because it is relatively easier to compute, allowing for numerical [19, 47, 48] and analytical [59, 60] studies of nonlocal magic in many-body quantum systems, particularly by applying tensor network methods. As the name suggests, the definition of mutual 2-SRE is directly reminiscent of mutual information of entanglement, and is thus expected to be free from UV divergences from the point of view of field theory [60]. Its connection to physical phenomena has also been revealed [19, 44, 48, 59, 60].

However, the mutual 2-SRE is still relatively expensive to compute for matrix product states (MPS), with the exact calculation scaling as $O(N\chi^{12})$ [16], where $\chi$ is the bond dimension of the MPS. Approximate contraction schemes and sampling-based techniques have also been proposed [19, 20, 61], with the latter mainly relying on Monte Carlo algorithms with local update rules. However, their convergence is not guaranteed, and their applicability needs to be tailored to the specific model under consideration. Overall, the existing approaches still face limitations in their general applicability and computational efficiency.

Here, we present an efficient characterization of various magic properties. First, we introduce the mutual von-Neumann SRE to characterize nonlocal magic where we find two possible definitions. For MPS, the mutual von-Neumann SRE can be efficiently calculated via perfect sampling in $O(N\chi^3)$ time, where $\chi$ is the bond dimension and $N$ the number of qubits. Additionally, we provide two new algorithms to compute the mutual 2-SRE based on Metropolis-Hastings Monte-Carlo. We show that mutual SREs are able to characterize the critical point of the transverse field Ising model (TFIM) in arbitrary local basis, overcoming the limitation of the bare SRE.

Further, we introduce the magic capacity $C_M$ in analogy to the entanglement capacity [62], and show that it is connected to the anti-flatness of the Pauli spectrum. We efficiently compute the capacity of magic for MPS in $O(N\chi^3)$ time, and show that it describes the sampling complexity to compute the von-Neumann SRE. We find that phases in Clifford+T circuits as well the ground state of the Heisenberg model are characterized by the scaling of the magic capacity with $N$. Notably, we find that magic capacity provides an efficient tool to discriminate between typical and atypical states.

Finally, we provide an improved algorithm to perform Pauli and Bell sampling, which we apply to compute the von-Neumann SRE and magic capacity for statevector simulations of arbitrary states. Our method scales as $O(8^{N/2})$ time and $O(2^N)$ memory (in contrast to naive sampling with $O(8^N)$ time and $O(4^N)$ memory) allow simulation of at least 24 qubits. Our work enhances the characterization of different aspects of magic of many-body quantum systems and quantum circuits.

The rest of the paper is structured as follows. We define general $\alpha$-SREs for mixed states in Sec. 2. Next, we introduce the mutual von-Neumann SRE and magic capacity in Sec. 3. In Sec. 4, we describe efficient numerical methods based on Pauli sampling, both for MPS and statevector simulations. In Sec. 5, we present our numerical results on both Clifford+T circuits and ground state phase transitions. Finally, we discuss our results in Sec. 6. In Table 1, we summarize the complexity of algorithms to compute SRE $M_\alpha$, mutual $\alpha$-SRE $\mathcal{I}_\alpha$ and magic capacity $C_M$ for MPS, statevector simulation and quantum computers.

Table 1: Complexity of computing SREs $M_\alpha$ and mutual SRE $\mathcal{I}_\alpha$ with Rényi index $\alpha$, as well as magic capacity $C_\mathrm{M}$. We consider algorithms for matrix product states (MPS) and statevector simulation. $N$ is the number of qubits, $\chi$ is the bond dimension of matrix product state, $\epsilon$ the additive accuracy.

| Index | Time |
|---|---|
| $\alpha$-SRE $M_\alpha$ via MPS | |
| $\alpha = 1$ | $O(N\chi^3 C_M \epsilon^{-2})$ [17, 18] |
| integer $\alpha > 1$ | $O(N\chi^{6\alpha})$ [16] |
| Mutual $\alpha$-SRE $\mathcal{I}_\alpha$ via MPS | |
| $\alpha = 1$ | $O(N\chi^3 C_M \epsilon^{-2})$ |
| $\alpha = 2$ | $O(N\chi^{12})$ [19] |
| Magic capacity $C_\mathrm{M}$ via MPS | |
| $C_\mathrm{M}$ | $O(N\chi^3 \epsilon^{-2})$ |
| $M_\alpha$ and $C_\mathrm{M}$ via statevector simulation | |
| any $\alpha$ | $O(8^N)$ [15] |
| $\alpha = 1$, $C_\mathrm{M}$ | $O(8^{N/2}\epsilon^{-2})$ |

## 2 SRE

For $N$-qubit pure states $|\psi\rangle$, magic can be measured using the SRE [15], defined as

$$M_\alpha(|\psi\rangle) = H_\alpha[p_{|\psi\rangle}] - N\ln 2 = \frac{1}{1-\alpha}\ln\left(2^{-N}\sum_{P\in\mathcal{P}_N}\langle\psi|P|\psi\rangle^{2\alpha}\right),$$

where $\mathcal{P}_N$ is the set of $4^N$ (unsigned) Pauli strings which are tensor products of single-qubit Pauli operators $\sigma^x$, $\sigma^z$, $\sigma^y$ and identity $I$. Here, we use the $\alpha$-Rényi entropy

$$H_\alpha[p] = \frac{1}{1-\alpha}\ln\left(\sum_p p^\alpha\right), \tag{1}$$

over the distribution of Pauli expectation values [15]

$$p_{|\psi\rangle}(P) = \frac{\langle\psi|P|\psi\rangle^2}{2^N}, \tag{2}$$

which satisfies $p_{|\psi\rangle} \geq 0$ and $\sum_{P\in\mathcal{P}_N} p_{|\psi\rangle}(P) = 1$. In the limit $\alpha \to 1$, one obtains the von-Neumann SRE [17, 18]

$$M_1(|\psi\rangle) = -2^{-N}\sum_{P\in\mathcal{P}_N}\langle\psi|P|\psi\rangle^2\ln\left(\langle\psi|P|\psi\rangle^2\right). \tag{3}$$

SREs have the following properties: i) For any pure stabilizer state $|\psi_\mathrm{C}\rangle$, $M_\alpha(|\psi_\mathrm{C}\rangle) = 0$, else $M_\alpha(|\psi\rangle) > 0$. ii) Invariant under Clifford unitaries $U_\mathrm{C}$, i.e. $M_\alpha(U_\mathrm{C}|\psi\rangle) = M_\alpha(|\psi\rangle)$, iii) Additive, i.e. $M_\alpha(|\psi\rangle\otimes|\phi\rangle) = M_\alpha(|\psi\rangle) + M_\alpha(|\phi\rangle)$. $M_\alpha$ is a monotone under channels that map pure states to pure states for $\alpha \geq 2$ [63], while this is not the case for $\alpha < 2$. Note that SREs are not strong monotones for any $\alpha$ [17], while they are for their linearized versions for $\alpha \geq 2$ [63].

In order to extend the SRE to mixed states, we need to generalize the probability distribution in Eq. (2). A modified distribution for mixed states can be constructed simply by adding a normalization factor, yielding

$$p_\rho(P) = 2^{-N} \frac{\text{tr}(\rho P)^2}{\text{tr}(\rho^2)}. \tag{4}$$

It is easy to see that $p_\rho \geq 0$ and $\sum_{P \in \mathcal{P}_N} p_\rho(P) = 1$. For pure states $p_{|\psi\rangle\langle\psi|} = p_{|\psi\rangle} = 2^{-N} \langle\psi| P |\psi\rangle^2$. We then define the $\alpha$-SREs for $N$-qubit mixed states $\rho$ from the viewpoint of entropies as

$$\tilde{M}_\alpha(\rho) = H_\alpha(p_\rho) + S_2(\rho) - N \ln(2), \tag{5}$$

where $S_2(\rho) = -\ln(\rho^2)$ is the 2-Rényi entropy. We note that the case $\alpha = 2$ is equivalent to the mixed-state SRE defined in Ref. [15]. If $\rho$ describes a pure state, $\rho = |\psi\rangle\langle\psi|$, then we have $\tilde{M}_\alpha(\rho) = M_\alpha(|\psi\rangle)$.

For mixed states, $\tilde{M}_\alpha(\rho)$ is zero for mixed stabilizer states of the form $\rho_{C_0} = \sum_{P \in G} \alpha_P P$ (i.e. states that have a pure stabilizer state as their purification) where $G$ is a commuting set of Pauli operators, and non-zero otherwise. However, states of the form of $\rho_{C_0}$ are only a subset of all mixed stabilizer states, which are convex mixtures of pure stabilizer states $|\psi_C^{(i)}\rangle$, i.e. $\rho_C = \sum_i x_i |\psi_C^{(i)}\rangle\langle\psi_C^{(i)}|$. Thus, for mixed states $\tilde{M}_\alpha(\rho)$ is not a proper magic monotone. Nevertheless, it has a nice property of being efficiently computable, allowing us to probe magic in subsystems of many-body states.

While the distribution $p_\rho$ appears to be the most natural extension, we will also consider an alternative probability distribution over Pauli strings $P$ as

$$q_\rho(P) = 2^{-N} \text{tr}(\rho P \rho P). \tag{6}$$

Note that this distribution is closely related to Bell sampling, which samples from $P \sim 2^{-N} \text{tr}(\rho^* P \rho P)$ [24, 64]. It is easy to see that $q_\rho(P) \geq 0$ due to $\rho \succeq 0$ and $P \rho P \succeq 0$. Further, we have $\sum_{P \in \mathcal{P}_N} q_\rho(P) = 1$ due to $\sum_{P \in \mathcal{P}_N} P \rho P = 2^N \text{tr}(\rho) I_n$ where $I_n$ is the identity. For pure states $q_{|\psi\rangle\langle\psi|} = p_{|\psi\rangle} = 2^{-N} \langle\psi| P |\psi\rangle^2$. A nice property of the distribution $q_\rho$ is that, for any subsystem $A$, the distribution for the reduced density matrix $\rho_A = \text{tr}_{A^c}(\rho)$ is simply the marginal distribution of $q_\rho$ in $A$. In other words,

$$q_{\rho_A}(P_A) = \sum_{P \in \mathcal{P}_{A^c}} q_\rho(P_A \otimes P), \tag{7}$$

for any $P_A \in \mathcal{P}_A$, which is proven in Appendix A. Here, given a subsystem $A$ with $N_A$ sites, $\mathcal{P}_A$ is the set of $4^{N_A}$ unsigned Pauli strings which consists of identities for any sites in the complement subsystem $A^c$. The property in Eq. (7) is particularly useful for computing the mutual SRE, which will be defined in Sec. 3.1. Note that the distribution $p_\rho$ does not possess an analogous property.

One can now also define mixed-state SREs using $q_\rho$. For $\alpha = 2$, this conincides with $\tilde{M}_2$, i.e.

$$\tilde{M}_2^{[q]}(\rho) = H_2[q_\rho] - S_2(\rho) - N \ln(2) \equiv \tilde{M}_2(\rho). \tag{8}$$

However, for $\alpha = 1$ one gains an alternate von-Neumann SRE

$$\tilde{M}_1^{[q]}(\rho) = H_1[q_\rho] - S_2(\rho) - N \ln(2). \tag{9}$$

where $\tilde{M}_1^{[q]} \neq \tilde{M}_1$ and $H_1[q] = -\sum_q q \ln(q)$. One can show that $\tilde{M}_1^{[q]}$ has the same properties as $\tilde{M}_1$, which one can show using $\tilde{M}_1^{[q]} \geq \tilde{M}_2 \geq 0$ from the hierarchy of Rényi entropies, and $\tilde{M}_1^{[q]}(\rho_C) = 0$.

# 3 Mutual magic and magic capacity

In this section, we introduce new magic quantities, including mutual $\alpha$-SRE $\mathcal{I}_\alpha$, an alternate definition of mutual von-Neumann SRE $\mathcal{I}_1^{[q]}$ and magic capacity $C_M$. We discuss how these quantities probe different aspects of magic. We will then show in the following sections that these quantities are efficiently computable with matrix product states.

## 3.1 Mutual SRE

We define mutual $\alpha$-SREs as

$$\mathcal{I}_\alpha(\rho) = \tilde{M}_\alpha(\rho) - \tilde{M}_\alpha(\rho_B) - \tilde{M}_\alpha(\rho_A), \tag{10}$$

where $A$ and $B$ are two subsystems of the state $\rho$. Note that Eq. (10) generalizes the definition for $\alpha = 2$ introduced in Ref. [19]. Note that in contrast to the usual mutual informations, $\mathcal{I}_\alpha$ can become negative, however $\mathcal{I}_2$ has still has found interesting applications in the context of many-body phenomena [19,44,48].

Using the alternate von-Neumann SRE defined in Eq. (9), we define an alternate version of the mutual von-Neumann SRE as

$$\mathcal{I}_1^{[q]}(\rho) = \tilde{M}_1^{[q]}(\rho) - \tilde{M}_1^{[q]}(\rho_A) - \tilde{M}_1^{[q]}(\rho_B), \tag{11}$$

which is equivalent to

$$\mathcal{I}_1^{[q]}(|\psi\rangle) = I_2(\rho) - I(q_{\rho_A}, q_{\rho_B}), \tag{12}$$

where

$$I_2(\rho) = S_2(\rho_A) + S_2(\rho_B) - S_2(\rho), \tag{13}$$

is the Rényi-2 quantum mutual information and

$$I(q_{\rho_A}, q_{\rho_B}) = H_1(q_{\rho_A}) + H_1(q_{\rho_B}) - H_1(q_\rho), \tag{14}$$

is the mutual information of the joint probability distribution $q_\rho$ defined in Eq. (6). Note that this relation holds because $q_{\rho_A}$ ($q_{\rho_B}$) is the marginal probability distribution of $q_\rho$ in $A$ ($B$). This provides $\mathcal{I}_1^{[q]}(|\psi\rangle)$ with a clear interpretation, as the difference between the quantum mutual information and the classical mutual information of $q_\rho$ defined in subsystems $A$ and $B$. Importantly, this also leads to a practical advantage in numerical simulations, as we will discuss in Sec. 4.1.

We find that $\mathcal{I}_2$, $\mathcal{I}_1$ and $\mathcal{I}_1^{[q]}$ behave similarly, though they can show different signs for the same state. We study them in Appendix B for the TFIM and Heisenberg model.

## 3.2 Magic capacity and anti-flatness of the Pauli spectrum

We now introduce the magic capacity in direct analogy to entanglement capacity [62] as

$$C_M(|\psi\rangle) = \text{var}(\hat{M}_1) = \mathbb{E}_{P \sim p(P)}[\ln(\langle\psi|P|\psi\rangle^2)^2] - M_1^2. \tag{15}$$

$C_M$ can be thought of as the variance of the estimator $\hat{M}_1$ to compute $M_1$ via Pauli sampling, and can be computed in $O(N\chi^3\epsilon^{-2})$ time for MPS. Note that, despite of the presence of constant shift in the definition of SRE, the expression for magic capacity is precisely equivalent to that used for entanglement capacity. Interestingly, entanglement capacity is closely related to the

anti-flatness of the entanglement spectrum, which has, in turn, been connected to magic [45, 53, 65]. If the explicit form of the SRE is known, the magic capacity can be calculated as [19]

$$C_M = \frac{\mathrm{d}^2\left[(1-\alpha)M_\alpha\right]}{\mathrm{d}\alpha^2}\bigg|_{\alpha=1}. \tag{16}$$

Similarly to entanglement capacity, the magic capacity is related to the anti-flatness of the Pauli spectrum. Specifically, magic capacity vanishes when the Pauli spectrum is flat, i.e., if the spectrum $p(P) = \text{const}$ for any Pauli string $P$ with non-zero expectation value.

As a toy example, consider a state whose Pauli spectrum is uniform with $\langle\psi|P|\psi\rangle^2 = 1/(2^N + 1)$ for all $P \in \mathcal{P}/\{I\}$. Although this spectrum does not correspond to a physical density matrix, it has been numerically observed to provide a sufficiently good approximation for the Pauli spectrum of a typical state [36]. It is clear that this spectrum is nearly flat, with only the identity operator deviating from a perfectly flat distribution. We can directly compute the SRE and the magic capacity for this uniform state, yielding

$$M_1^{\mathrm{uniform}} = (1 - 2^{-N})\ln\!\left(2^N + 1\right), \tag{17}$$

and

$$C_M^{\mathrm{uniform}} = 2^{-N}(1 - 2^{-N})\ln^2(2^N + 1). \tag{18}$$

In the limit $N \to \infty$, we have $M_1^{\mathrm{uniform}} = N\ln(2)$ and $C_M^{\mathrm{uniform}} \to 0$. The vanishing magic capacity captures the intuition that the Pauli spectrum of the uniform state is asymptotically flat, as expected. We compute the magic capacity of typical states in Appendix C, finding similar scaling with $C_M = \text{const}$, while product states have $C_M \propto N$.

Note that, since the Pauli spectrum always includes the identity operator, $p(I) = 2^{-N}$, it follows that only stabilizer states exhibit flat Pauli spectrum. In other words, magic capacity vanishes if and only if the state is a pure stabilizer state. In fact, magic capacity satisfies similar properties as the SREs: i) For any pure stabilizer state $|\psi_C\rangle$, $C_M(|\psi_C\rangle) = 0$, else $C_M(|\psi\rangle) > 0$. ii) Invariant under Clifford unitaries $U_C$, i.e. $C_M(U_C\rho U_C^\dagger) = C_M(\rho)$, iii) Additive, i.e. $C_M(\rho \otimes \sigma) = C_M(\rho) + C_M(\sigma)$. Thus, we may also view magic capacity as a measure of magic. However, magic capacity captures a different type of complexity, namely the complexity of estimating the magic measure itself. This is different from the conventional notion of a measure of magic, which typically characterizes the hardness of classical simulation by Clifford-based techniques or the number of non-Clifford gates required to prepare the state.

To illustrate this point further, let us consider the estimation of $M_\alpha$ for $\alpha \neq 1$. In this case, the SRE can be estimated by

$$M_\alpha(|\psi\rangle) = \frac{1}{1-\alpha}\ln \mathop{\mathbb{E}}_{P\sim p(P)}\left[\langle\psi|P|\psi\rangle^{2(\alpha-1)}\right], \tag{19}$$

and the variance of this estimator is given by [19]

$$\mathrm{Var}(M_\alpha) \approx \frac{\exp\left[2(\alpha-1)(M_\alpha - M_{2\alpha-1})\right] - 1}{|\alpha-1|}. \tag{20}$$

We see that the variance is dictated by $M_\alpha - M_{2\alpha-1}$. This difference between SREs of different Rényi indices also serves as a measure of the anti-flatness of the Pauli spectrum, analogous to the anti-flatness measures for the entanglement spectrum [53]. This highlights the direct connection between anti-flatness of the Pauli spectrum and the complexity of estimating magic from samples of Pauli operators.

# 4 Efficient numerical methods by Pauli sampling

In this section, we present several algorithms to compute the SREs, mutual von-Neumann SRE, and magic capacity, all of which are based on sampling of Pauli strings.

## 4.1 Perfect sampling of mutual von-Neumann SRE for MPS

The von-Neumann SRE $M_1(|\psi\rangle)$ for pure MPS $|\psi\rangle$ can be efficiently computed via perfect sampling as shown in Ref. [17, 18], where the complexity scales as $O(N C_M \chi^3 \epsilon^{-2})$, where $\epsilon$ is the additive precision. In particular, the error $\epsilon$ to estimate $M_1$ scales as

$$\epsilon \sim \sqrt{C_M/K}, \tag{21}$$

where magic capacity $C_M = \text{var}(\hat{M}_1)$ relates the variance of $m_1$ over the perfect sampling protocol and is upper bounded by $C_M = O(N^2)$ [18] and $K$ is the number of samples. Notably, $C_M$ can have widely different scaling with $N$ depending on the model and parameter regime. Similarly, $C_M$ as defined in Eq. (15) can be estimated via perfect Pauli sampling [17, 18] similarly to $M_1$.

The mutual von-Neumann SRE $\mathcal{I}_1^{[p]}(|\psi\rangle)$ can be evaluated efficiently using the methods of Ref. [18]. In particular, one has to sample from the normalized Pauli distribution $p_\rho$ as given by Eq. (4), which can be done efficiently as long as $\rho$ has a purification in form of an MPS. The SREs for each of the subsystems $A, B$, and $AB$ in Eq. (10) can then be evaluated separately in an efficient way. Note that, typically the leading extensive term in the SRE is cancelled out, leaving a subleading term that is much smaller than the SRE. As a result, the estimation of $\mathcal{I}_1^{[p]}(|\psi\rangle)$ is expected to require a significantly larger number of samples compared to the typical estimation of $M_1$ (see Appendix B).

While a similar approach can be used to compute the alternate version of the mutual von-Neumann SRE $\mathcal{I}_1^{[q]}(|\psi\rangle)$, here we give an even more efficient way to compute $\mathcal{I}_1^{[q]}(|\psi\rangle)$. Using the expression in Eq. (12), we can write

$$\mathcal{I}_1^{[q]}(|\psi\rangle) = S_2(\rho_A) + S_2(\rho_B) - I(q_{\rho_A}, q_{\rho_B}), \tag{22}$$

where $I(q_{\rho_A}, q_{\rho_B})$ is defined in Eq. (14). The 2-Rényi entanglement entropies can be efficiently computed for MPS via standard methods. Then, $I(q_{\rho_A}, q_{\rho_B})$ can be estimated by

$$I(q_{\rho_A}, q_{\rho_B}) = \mathop{\mathbb{E}}_{P \sim q(P)} [\ln(\text{tr}(\rho P \rho P)) - \ln(\text{tr}(\rho_A P_A \rho_A P_A)) - \ln(\text{tr}(\rho_B P_B \rho_B P_B))],$$

where $P_A$ and $P_B$ are Pauli strings reduced onto subsystem $A$ and $B$ respectively, such that $P = P_A \otimes P_B$. Thus, we need to sample only from one distribution $q_\rho$, which can be done efficiently for MPS as shown in Ref. [17]. We discuss the technical details in Appendix D. Importantly, the estimation of $\mathcal{I}_1^{[q]}$ is expected to require significantly fewer samples than $\mathcal{I}_1^{[p]}$ because it avoids the need to accurately estimate the individual SREs for the subsystems. We corroborate this computational advantage in the transverse-field Ising model, as shown in Appendix B.

## 4.2 Monte Carlo algorithms for mutual 2-SRE

Next, we introduce improved methods to compute the mutual SRE $\mathcal{I}_2$ for $\alpha = 2$. Previous works have estimated $\mathcal{I}_2$ by Monte Carlo sampling of Pauli strings in tensor networks [19, 47, 48, 61]. To do this, one rewrites Eq. (10) as follows:

$$\mathcal{I}_2(\rho) = I_2(\rho) - B(\rho), \tag{23}$$

where

$$B(\rho) = -\ln\left(\frac{\sum_{P_A\in\mathcal{P}_A}|\operatorname{Tr}(\rho_A P_A)|^4 \sum_{P\in\mathcal{P}_B}|\operatorname{Tr}(\rho_B P_B)|^4}{\sum_{P\in\mathcal{P}}|\operatorname{Tr}(\rho P)|^4}\right), \tag{24}$$

and 2-Rényi mutual information $I_2(\rho) = S_2(\rho_A) + S_2(\rho_B) - S_2(\rho)$. In the case of complementary subsystems, the 2-Rényi mutual information is simply given by the 2-Rényi entanglement entropy, which is readily accessible in MPS. To estimate $B(\rho)$, one can sample the Pauli strings according to $\Pi(P) \propto \operatorname{Tr}(\rho P)^4$ and compute

$$B(\rho) = -\ln \mathop{\mathbb{E}}_{P\sim\Pi(P)}\left[\frac{|\operatorname{Tr}(\rho_A P_A)|^4 |\operatorname{Tr}(\rho_B P_B)|^4}{|\operatorname{Tr}(\rho P)|^4}\right], \tag{25}$$

where $P$ is decomposed as $P = P_A \otimes P_B$. Since expectation values of Pauli strings can be efficiently computed in MPS, one can perform Metropolis Monte Carlo scheme by locally updating Pauli string configurations.

The limitation of such an approach is that one must deal with equilibration and autocorrelation times, leading to reduced efficiency. Moreover, the local update scheme need to be tailored to the specific model being studied. In particular, the presence of symmetry may necessitate the design of intricate multi-site update schemes to ensure ergodicity of the Markov chain. While a perfect sampling scheme is able to solve these issues, it cannot be directly applied to estimate $B(\rho)$ since it can only sample from the Pauli distribution $p(P)$.

To mitigate these problems, we propose a method that combines the Monte Carlo and perfect sampling techniques. Specifically, we employ the Metropolis-Hastings Monte Carlo algorithm [66] to sample from $\Pi(P)$. In this scheme, given a current configuration $P$, a candidate configuration $P'$ is proposed according to some prior distribution $g(P'|P)$, which we assume we can draw from. The candidate is then accepted with probability

$$P_{\mathrm{acc}}(P \to P') = \min\left\{1, \frac{g(P|P')}{g(P'|P)}\frac{\Pi(P')}{\Pi(P)}\right\}. \tag{26}$$

The ideal prior is $g(P'|P) = \Pi(P')$, which is however not available through direct sampling. In Metropolis Monte Carlo, $P'$ is generated by locally updating $P$ and the corresponding $g(P'|P)$ is symmetric in its arguments. Here, we propose different types of prior that approximate the ideal prior $\Pi(P')$, and which can be directly sampled in MPS. These approaches also have the advantage that all samples are independent from each other, and thus can be generated in parallel.

First, we propose to use $p(P')$ as a prior, which is readily available through perfect sampling. Given that $p(P')$ and $\Pi(P')$ correspond to different powers of Pauli strings, we expect that these probability distributions are strongly correlated. This suggests that employing $p(P')$ as the proposal distribution within the Metropolis-Hastings framework could improve sampling efficiency. This intuition is particularly clear for stabilizer states, in which case $p(P)$ and $\Pi(P)$ are identical. More generally, the difference between two probability distributions can be quantified using the Kullback-Leibler divergence, defined as

$$D_{\mathrm{KL}}(P(x)\|Q(x)) = \sum_x P(x)\ln\frac{P(x)}{Q(x)}, \tag{27}$$

where $P(x)$ and $Q(x)$ are probability distributions. We find that the Kullback-Leibler divergence between $p(P)$ and $\Pi(P)$ is related to the difference of SRE, as follows

$$D_{\mathrm{KL}}(p(P)\|\Pi(P)) = M_1 - M_2. \tag{28}$$

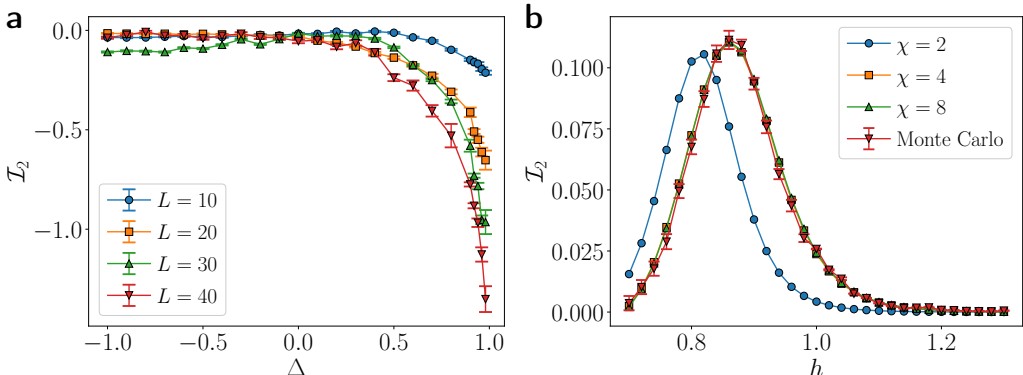

Figure 1: Mutual 2-SRE $\mathcal{I}_2$ of the groundstate of the (a) the Heisenberg model with bond dimension $\chi = 20$, and (b) TFIM with $N = 16$. For the TFIM, the Monte Carlo results are obtained with $\chi = 8$ and $\chi' = 2$. The mutual SRE is calculated in respect to two subsystems $A, B$ of size $N/4$ located at the respective boundaries of the chain.

As discussed in Sec. 3.2, such a difference of SRE characterizes the anti-flatness of Pauli spectrum. This approach thus works best for states with a nearly flat Pauli spectrum. In the case where the spectrum is far from being flat, exponentially many samples may be required for convergence to the desired distribution. We note that a similar exponential sample problem also arises in estimating the 2-SRE through sampling techniques [17, 18]. Nevertheless, the overall cost typically grows much more slowly than that of exact computation [19, 61]. Therefore, even with the potential for high sample complexity in some cases, this approach remains valuable for extending the accessible system sizes.

We briefly illustrate this method for the ground-state of the XXZ or Heisenberg model which is given by

$$H_{\text{XXZ}} = -\sum_{n=1}^{N-1} \left( \sigma_n^x \sigma_{n+1}^x + \sigma_n^y \sigma_{n+1}^y + \Delta \sigma_n^z \sigma_{n+1}^z \right), \tag{29}$$

with the anisotropy $\Delta$. The model conserves the total number of excitations. We compute the ground state within the half-filling excitation symmetry sector $N_{\text{p}} = \sum_{n=1}^{N} \sigma_n^z = 0$. In Fig. 1a, we show the mutual 2-SRE $\mathcal{I}_2$ as a function of the field $\Delta$. It has a similar qualitative behavior as the mutual von-Neumann SRE. Further, we observed a significant increase in the statistical errors as the anisotropy approaches $\Delta = 1$. To maintain data quality, we thus required a substantially larger number of samples $K = 10^8$ for $\Delta \geq 0.9$ compared to $K = 10^6$ for $\Delta < 0.9$. This can be seen as a consequence that the anti-flatness of the ground states drastically increases as the anisotropy approaches $\Delta = 1$, as we will demonstrate in Sec. 5 using the magic capacity $C_M$.

Another strategy to approximate the probability distribution $\Pi(P)$ is to construct an MPS approximation $\left| \text{MPS}' \right\rangle$ of the target MPS by truncating the bond dimension $\chi$ to a smaller bond dimension $\chi'$. The distribution $\tilde{\Pi}(P) \propto \left\langle \text{MPS}' \right| P \left| \text{MPS}' \right\rangle^4$ can be used as an approximation to $\Pi(P)$. For sufficiently small $\chi'$, $\tilde{\Pi}(P)$ can be sampled directly by exact contraction of the MPS with a computational cost scaling as $O(\chi'^9)$. This allows us to implement the Metropolis-Hastings scheme by selecting the prior $g(P'|P) = \tilde{\Pi}(P)$. Note that this approach also enables the estimation of mutual magic when the subsystems are separated, a scenario that has found importance in physical phenomena [19, 48, 59]. In Fig. 1b, we compare the mutual SRE obtained by exact contraction with the Monte Carlo estimation for $N = 16$ for the ground state of the TFIM model

$$H_{\text{TFIM}} = -\sum_{k=1}^{N-1} \sigma_k^x \sigma_{k+1}^x - h \sum_{k=1}^{N} \sigma_k^z, \tag{30}$$

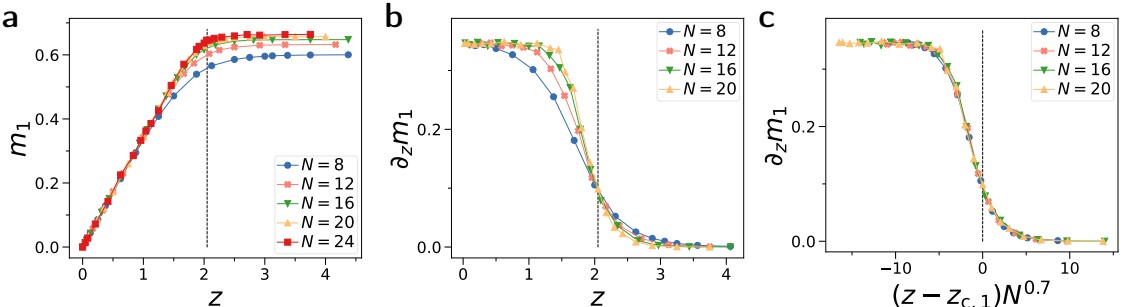

Figure 2: Von-Neumann SRE density $m_1$ for Clifford circuits doped with T-gate density $z = N_T/N$. **a)** Average $m_1$ against $z$ for different qubit numbers $N$. **b)** $\partial_z m_1$ against $z$. Curves of $\partial_z m_1$ intersect for all $N$ at saturation transition $z_{c,1}^{fit} \approx 2.05$, which is marked by the dashed black line. **c)** $\partial_z m_1$ against rescaled $(z - z_{c,1}^{fit})N^\gamma$, where we find scaling factor $\gamma \approx 0.7$. After rescaling, the curves collapse for all $N$, showing the universality of $\partial_z m_1$.

where we have the field $h$ and choose open boundary conditions. Here, $A, B$ are two subsystems located at the boundary of the chain, each of length $N/4$. For the Monte Carlo estimation, we use $\chi = 8$ and $\chi' = 2$. While the MPS $\chi' = 2$ approximation exhibits deviations from the exact mutual SRE, the samples generated from this approximation can still serve as effective proposals within the Metropolis-Hastings framework. This approach leads to accurate Monte Carlo estimation of the mutual SRE, demonstrating the efficacy of our method.

### 4.3 Statevector simulation for Pauli sampling, magic capacity and SRE

Finally, we introduce a statevector simulation method for Pauli sampling, Bell sampling, magic capacity, and von-Neumann SRE with time complexity $O(2^{N/2})$ and memory $O(2^N)$.

Pauli sampling involves sampling from the Pauli distribution $P \sim p_{|\psi\rangle}(P) = 2^{-N} \langle\psi| P |\psi\rangle^2$. It has for example applications in direct estimation of fidelity [67]. Pauli sampling is closely related to Bell sampling $P \sim 2^{-N} \langle\psi^*| P |\psi\rangle^2$, which can be efficiently done in experiment [25, 64,68–70] with many applications, such as learning stabilizer states [64] with few T-gates [70–72] and estimating magic [24, 25]. In fact, our Pauli sampling algorithm can also be used to perform Bell sampling.

From Pauli sampling, one can efficiently compute von-Neumann SREs and magic capacity. Simulating Pauli sampling naively on a classical computer requires storing a $4^N$ dimensional state in memory, which in general requires too much memory to be practical. To reduce memory load, one could consider Feynmann type simulation algorithms which have lower memory cost, however the trade-off is an excessive computational time [73].

To balance memory and computational time, we provide a hybrid Schrödinger-Feynmann algorithm to sample Pauli $P$ from the distribution $p(P)$, where each sample takes $O(8^{N/2})$ time and $O(2^N)$ memory. The algorithm is presented in detail in Appendix E The main idea of our algorithm is to sample the $2N$ bits of the Pauli in two steps: The first $N$ qubits are sampled with a Feynmann-like algorithm where the state is not explicitly stored in memory. Then, after measuring $N$ qubits the reduced state can be stored fully in memory and the outcomes for the final $N$ qubits are sampled via marginals.

From this, we get an improved method to compute SREs. A naive method to compute SREs is to sum over exponentially many Pauli expectation values $\langle\psi| \sigma |\psi\rangle^{2\alpha}$. This method scales as $O(8^N)$ and has been implemented for up to 15 qubits. With Pauli sampling, we achieve a square-root speedup to compute the case $\alpha = 1$, allowing us to compute von-Neumann SREs for 24 qubits. Similarly, we can compute the magic capacity via the Pauli sampling.

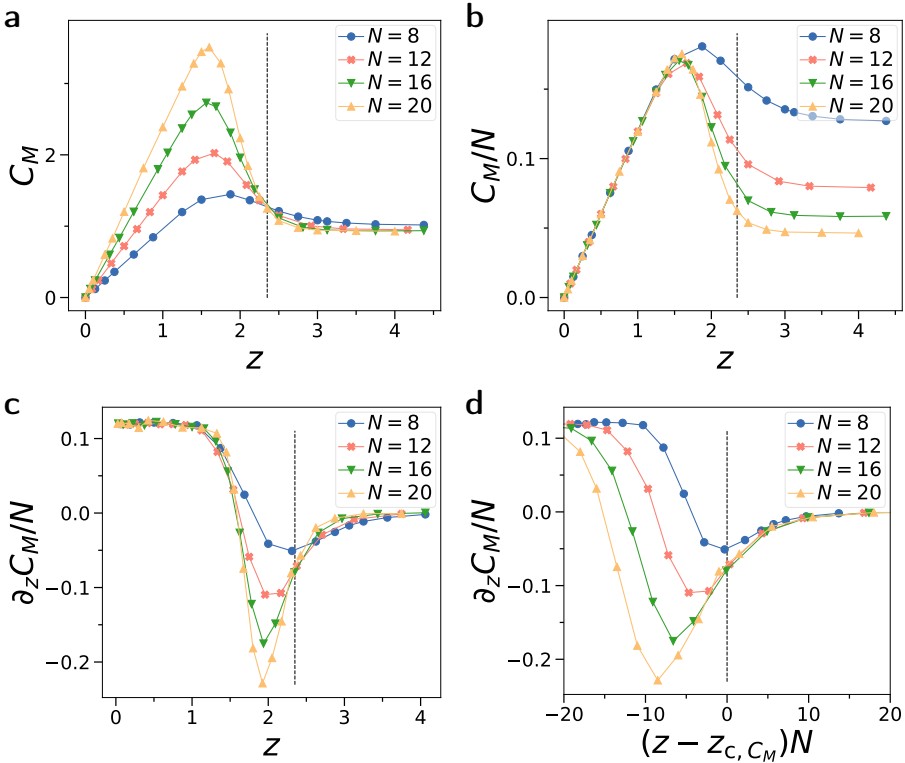

Figure 3: Magic capacity $C_\mathrm{M}$ for Clifford circuits doped with T-gate density $z = N_\mathrm{T}/N$. **a)** Magic capacity $C_\mathrm{M}$ against $z$. Curves for all $N$ intersect at a transition point $z_{\mathrm{c},C_\mathrm{M}} \approx 2.35$, which is marked by vertical dashed line. We used $K = 10^5$ Pauli samples for $N = 8$ and $N = 12$, while $K = 10^3$ for $N = 16$ and $N = 20$. Data is averaged over at least 100 random instances. **b)** Magic capacity density $C_\mathrm{M}/N$, which is rescaled with $1/N$. **c)** Derivative of magic capacity density $\partial_z C_\mathrm{M}/N$ against $z$. Curves for $N \geq 12$ intersect at transition point $z_{\mathrm{c},C_\mathrm{M}} \approx 2.35$. **b)** $\partial_z C_\mathrm{M}/N$ against rescaled $(z - z_{\mathrm{c},C_\mathrm{M}})N$. Around the transition point $z_{\mathrm{c},C_\mathrm{M}}$, curves for $N \geq 12$ can be made to collapse to a single curve.

The detailed algorithm is presented in Appendix F where the code is available on Github from Ref. [74]. Our algorithm gives us an unbiased estimator of $M_1(|\psi\rangle)$, where we can bound the scaling of the estimation error as $\epsilon \sim \sqrt{C_M/K}$, where $K$ is the number of measured samples. We achieve a scaling of a $O(8^{N/2}\epsilon^{-2})$ in time and $O(2^N)$ in memory. Our algorithm has the same asymptotic scaling as the perfect sampling algorithm for MPS with maximal bond dimension $\chi = 2^{N/2}$, but does not need the explicit construction of the MPS, which is convenient for many applications.

## 5 Applications to many-body phenomena

We apply the methods in Sec. 4 to investigate the mutual von Neumann SRE and magic capacity, both in the context of ground states and quantum circuits. For the Clifford+T circuits, we perform the statevector simulation in Sec. 4.3. For ground states, we obtain them using DMRG, and then employ the MPS Pauli sampling method.

## 5.1 Clifford+T circuits

First, we study $m_1$ and magic capacity $C_M$ to characterize random Clifford circuits doped with T-gates. They are random Clifford circuits $U_C$ interleaved with $N_T$ T-gates $T = \text{diag}(1, \exp(-i\pi/4))$ [25, 75]

$$|\psi(N_T)\rangle = U_C^{(0)} \Bigg[ \prod_{k=1}^{N_T} (T \otimes I_{n-1}) U_C^{(k)} \Bigg] |0\rangle . \tag{31}$$

Here, we define the T-gate density $z = N_T/N$. It is known that Clifford+T circuits with $z = \text{const}$ become close approximations of Haar random states [34, 75–77], where the exact transition point is not known. Further, it has been observed that Clifford+T circuits saturate the maximal possible value of the SRE $M_\alpha$ in a sharp transition for large $N$. Such saturation transitions appear at a critical $z_{c,\alpha}$, which depends on the chosen $\alpha$ of the $\alpha$-SRE [36]. The saturation transition is characterized by a universal behavior of $\partial_z m_\alpha$, where $m_\alpha = M_\alpha/N$ is the magic density and $\partial_z$ is the derivative in respect to $z$. In particular, one finds that the curves of $\partial_z m_\alpha$ for different $N$ cross at the same $z_{c,\alpha}$. Further, by a proper rescaling, $\partial_z m_\alpha(N)$ can be made to collapse to a single curve for all $N$ [36]. This is hallmark of universal behavior, usually found at phase transitions, where properties of the system become scale-invariant and simple functions of $N$ [78]. The transition points have been found analytically, e.g. $z_{c,0}^{\text{analytical}} = 1$, $z_{c,1}^{\text{analytical}} = 2$ and $z_{c,2}^{\text{analytical}} = \ln(2)/\ln(4/3) \approx 2.409$ [36]. However, the functional shape of the curves $M_\alpha(N)$ around the transition is known analytically only for $\alpha = 2$, while for other $\alpha$ numerical studies for sufficiently large $N$ have been challenging. As Clifford+T circuits are highly entangled quantum states, MPS techniques are not suitable for their simulations. Instead, we use our method for statevector simulation introduced in Sec. 4.3 to compute $M_1$ and $C_M$ up to $N = 24$ qubits.

In Fig.2a, we show the von-Neumann SRE density $m_1 = M_1/N$ against T-gate density $z = N_T/N$. We find that it increases linearly with $z$ until it saturates at sufficiently large $z$. For large $N$, this transition to maximal $m_1$ is expected to become sharp, at a characteristic saturation transition point $z_{c,1}$ [36]. In Fig.2b, we plot the derivative $\partial_z m_1$ against $z$. We find that the curves for all $N$ intersect at $z_{c,1}^{\text{fit}} \approx 2.05$ marked with a dashed black line, which indicates the critical point of the saturation transition. Note that the numeric value matches closely the analytic value of $z_{c,1}^{\text{analytic}} = 2$ derived in Ref. [36] for $\alpha = 1$. Notably, around the critical point, the curves become universal, i.e. they follow a functional form that has a simple dependence on $N$. In particular, by rescaling $(z - z_{c,1}^{\text{fit}})N^\gamma$ with $\gamma \approx 0.7$, we find that all curves clearly overlap in Fig.2c.

Next, we study magic capacity $C_M$ in Fig.3, where as we will see observe the transition at a different $z_{c,C_M}$. In Fig.3a, we show the magic capacity $C_M$ against $z$. We find that it increases with $z$, until it peaks and decreases sharply onto a constant, nearly $N$-independent value. Notably, for large $N$ and $z$, the value found matches the typical $C_M^{\text{typ}} \approx 0.9348$ for Haar random states derived in Eq. (C.3), indicating that at $z_{c,C_M}$ Clifford+T circuits become typical and are good approximations of Haar random states, quantifying predictions of Refs. [34, 75]. Notably, we see that at $z_{c,C_M} \approx 2.35$, curves for different $N$ intersect at a single point which is indicated by the vertical black line. This indicates a transition in $C_M$ at this point, which is notably at a different point than $z_{c,1}$. In Fig.3b, we study the magic capacity density $C_M/N$. For $z \ll z_{c,C_M}$, we find that $C_M \propto N$, indicating that the capacity increases with $N$, in contrast to $z > z_{c,C_M}$. Next, in Fig.3c we study the derivative of the magic capacity density $\partial_z C_M/N$. We find that curves for $N \geq 12$ intersect at $z_{c,C_M}$. We note that small $N = 8$ does not intersect at the saturation transition point due to finite size effects. Further, in Fig.3d, we plot the derivative against rescaled $(z - z_{c,C_M})N$. Here, we find that around $z_{c,C_M}$, the curves coincide for different

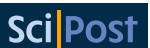

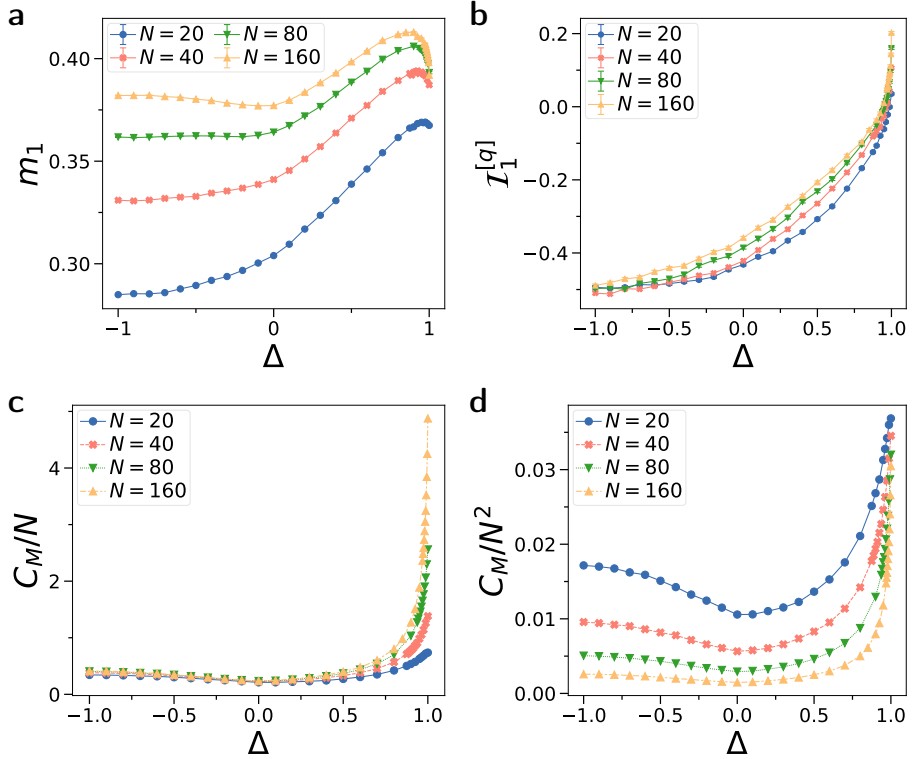

Figure 4: Von-Neumann SRE for groundstate of Heisenberg model with anisotropy $\Delta$. **a)** $m_1 = M_1/N$, **b)** mutual von-Neumann SRE $\mathcal{I}_1^{[q]}$ for bipartitions of size $N/2$. **c)** Magic capacity density $C_M/N$ against $\Delta$. **d)** Magic capacity $C_M$ rescaled with $1/N^2$. We use $K = 10^5$ samples to estimate $m_1$ and $C_M$.

$N$, for $N \geq 12$. This highlights that $C_M$ also undergoes a saturation transition similar to $m_\alpha$, however the transition happens at larger $z$.

Finally, we note that the saturation transition in magic capacity $z_{c,C_M}$ implies a transition in the number of samples $K$ needed to estimate $M_1$. In particular, the error scales as $\epsilon \sim \sqrt{C_M/K}$ as $C_M$ determines the variance of the estimator of $M_1$ (see Appendix F). In particular, the variance determines the number of sampled Pauli strings needed to estimate $M_1$ within a given accuracy. Thus, we find that the von-Neumann SRE requires the most samples for $z \approx 1.7$ where $C_M$ shows a peak and scales as $C_M \propto N$. In contrast, for $z > z_{c,C_M}$, $C_M$ is small and independent of $N$, rendering the estimation of the SRE substantially easier when states become typical.

## 5.2 Hamiltonian ground state transitions

Next, we use the magic capacity and mutual magic to characterize transitions in the ground state of Hamiltonians.

First, we study the Heisenberg model as defined in Eq. (29) as function of anisotropy $\Delta$. We show $m_1$ in Fig. 4a and the mutual SRE in Fig. 4b. Notably, we find that towards $\Delta = 1$, $m_1$ develops a minima, while $\mathcal{I}_1^{[q]}$ acquires a maximum. Next, we look at the magic capacity $C_M$ (and thus variance of the estimator of $M_1$) in Fig. 4c,d. We find that crucial information resides in the scaling of $C_M$ with $N$ which we study by plotting $C_M$ rescaled with powers of $N$. In Fig. 4c, we find that for $\Delta \ll 1$ we have $C_M \propto N$. In contrast, we find in Fig. 4d $\Delta \approx 1$, the magic capacity scales as $C_M \propto N^2$. Notably, this asymptotically saturates the scaling, as is upper bounded by $C_M = O(N^2)$. Note that this demonstrates the $\Delta = 1$ ground state is a

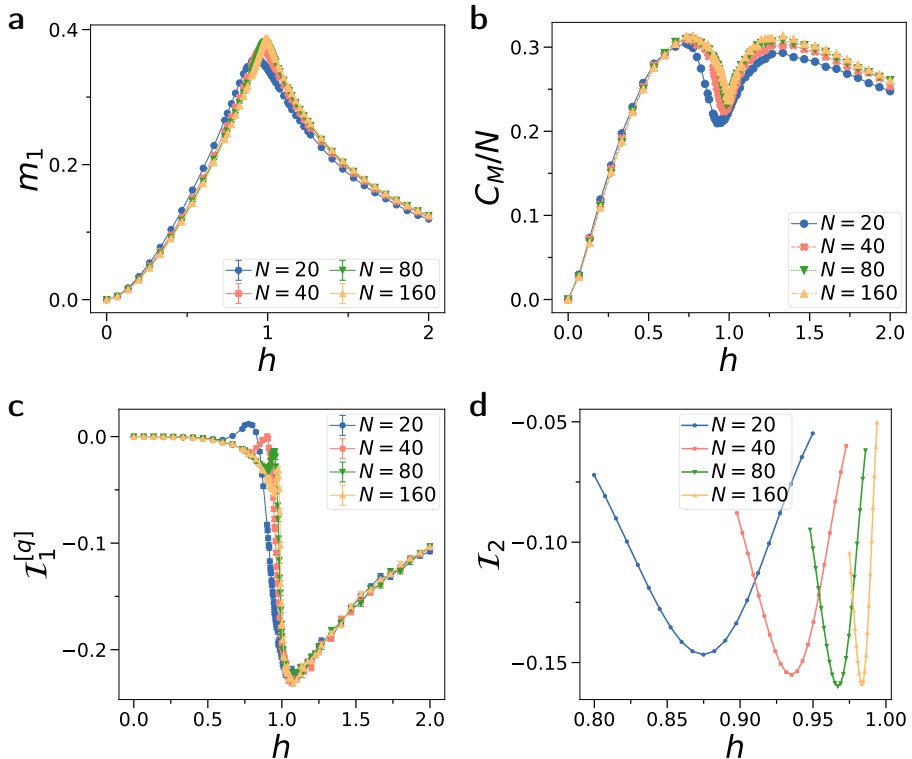

Figure 5: Von-Neumann SRE for groundstate of TFIM against field $h$, where we choose the Hamiltonian in the standard computational basis. **a)** $m_1 = M_1/N$ and **b)** $C_M$ rescaled by $1/N$. **c)** Mutual von-Neumann SRE $\mathcal{I}_1^{[q]}$ and **d)** Mutual 2-SRE for bipartitions of size $N/2$. We use $K = 10^5$ Pauli samples to estimate $m_1$, $C_M$ and $\mathcal{I}_1^{[q]}$.

highly atypical state, which shows a quite distinct scaling compared to Haar random states (with $C_M = O(1)$) and product states (with $C_M \propto N$) as shown in Appendix C.

Next, in Fig. 5 we study the ground state of the TFIM and show the von-Neumann SRE density $m_1 = M_1/N$ in Fig. 5a for different $N$. $m_1$ peaks close to the critical point $h \approx 1$ and converges to $h = 1$ for increasing $N$. Similarly, in Fig. 5b, we study the magic capacity $C_M$. We find that $C_M \propto N$ for all $h$, and exhibits a minimum at the critical point $h = 1$, indicating that the magic capacity is also sensitive to criticality. Then, we show the mutual von-Neumann SRE $\mathcal{I}_1^{[q]}$ in Fig. 5c and 2-SRE $\mathcal{I}_2$ in Fig. 5d. We find that both shows a sharp peak at the critical point $h = 1$. Notably, both $\mathcal{I}_2$ and $\mathcal{I}_1^{[q]}$ do not increase substantially with $N$ at the critical point, highlighting that there is no logarithmic divergence and only constant corrections with $N$. We can use the minima of $C_M$ and $\mathcal{I}_2$, as well as the peaks of $m_1$ and $\mathcal{I}_1^{[q]}$ to determine the critical field of the TFIM. We adopt the fitting procedure of Ref. [16], which used $m_2$. Here, we track the field $h_0(N)$ with extremal value of $m_1$, $C_M$, $\mathcal{I}_2$ and $\mathcal{I}_1^{[q]}$ for different $N$, and then extrapolate the fit to $N \to \infty$ to determine $h_c^{\text{fit}} \equiv h_0(N \to \infty)$. We demonstrate this procedure in Appendix G, where we find accurate match between true critical field $h_c = 1$ and our fits $h_c^{\text{fit}}$.

The mutual SRE is also highly useful to determine the critical point of the TFIM independent of the chosen local basis. The amount of magic changes when the local basis of the Hamiltonian is rotated as $H' = V^{\otimes N} H V^{\otimes N\dagger}$ with single-qubit unitary $V$. As a result, depending on $V$, there may be no extremum in $M_\alpha$ at the critical point [16]. In contrast, we find that mutual SREs such as $\mathcal{I}_1^{[q]}$ or $\mathcal{I}_2$ show a peak at $h = 1$ independent of the choice of $V$. We perform a scaling analysis for $\mathcal{I}_2$ in Appendix H, where we find via fitting a critical point $h_c^{\text{fit}} \approx 1.004$.

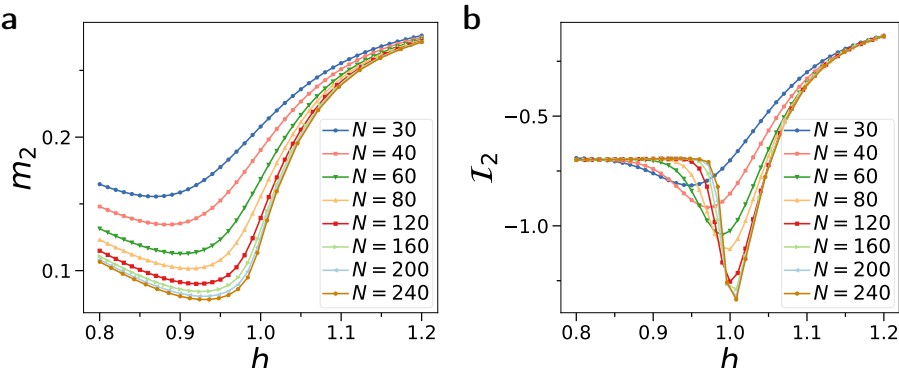

Figure 6: 2-SRE for groundstate of TFIM with field $h$ in a locally rotated basis $V_y = \exp(-i\sigma^y \pi/8)$ for different $N$ to determine critical point $h = 1$. We show **a)** $m_2$ and **b)** $\mathcal{I}_2$.

We highlight this for the choice $V_y = \exp(-i\sigma^y \pi/8)$ in Fig. 6. While in Fig. 6a $m_2$ does not show an extremum at the critical point (the observed minimum does not converge towards $h = 1$ even for large $N$ [16]), a minimum at $h = 1$ clearly emerges for $\mathcal{I}_2$ in Fig. 6b. We study the criticality via mutual SRE further in Appendix H for different basis and types of mutual SRE.

Finally, we note that recent conformal field theory (CFT) calculations [60] reveal that the term $B(\rho)$ in Eq. (23) displays a universal logarithmic scaling. In Ising CFT, the logarithmic scaling has the same prefactor as the 2-Rényi mutual information $I_2$. As a result, the logarithmic scaling term cancels out in the mutual 2-SRE, leaving a constant term that is likely non-universal. Our data is consistent with the CFT prediction, as we find no logarithmic divergence in the standard basis (i.e. Fig. 5). Nevertheless, the mutual SRE still displays non-analytic behavior at the critical point in this case.

## 6 Discussion

In this work, we present several advancements in the efficient estimation and characterization of magic in many-body quantum systems and quantum circuits.

First, we introduced the mutual von Neumann SRE as a proxy of nonlocal magic, which can be efficiently computed using perfect Pauli sampling in $O(N\chi^3)$ time for MPS. We find two different variants $\mathcal{I}_1$ and $\mathcal{I}_1^{[q]}$, where for the latter we provide a simpler algorithm for computation. Moreover, we develop two new algorithms based on Metropolis-Hastings Monte Carlo methods to compute the mutual 2-SRE. Remarkably, our analysis reveals that mutual SREs consistently identify critical points in the TFIM regardless of the local basis chosen. This contrasts sharply with the limitations of the bare SRE, which fails to detect criticality under basis rotations [16]. This demonstrates the enhanced robustness of mutual SREs for characterizing quantum phase transitions, independent of the local basis.

Curiously, we find that the mutual SRE does not exhibit logarithmic divergence at the critical point in the natural basis, which is also in agreement with the CFT prediction [60]. This behavior is in contrast to the findings of previous works [44, 61], which found logarithmic scaling of mutual magic in critical ground states [61] and monitored circuits [44]. This difference is possibly due to the fact that the latter two works employed genuine measures of magic for mixed states to define the mutual magic. This calls into question about the role of mutual SRE as a proxy of nonlocal magic. The efficient methods to compute mutual SRE we developed in this work would enable systematic investigation of the mutual SRE, paving the way for a deeper understanding on its relation to nonlocal magic.



Further, we define the magic capacity $C_M$, in analogy to entanglement capacity, and discuss its connection to the anti-flatness of the Pauli spectrum. This parallels the relationship between entanglement capacity and the anti-flatness of the entanglement spectrum. We show that magic capacity can be viewed as a measure of magic, which however probes a different type of complexity. Namely, it characterizes the sampling complexity to compute the von-Neumann SRE. Importantly, magic capacity can also be efficiently computed for MPS in $O(N\chi^3)$ time.

Using $C_M$, we can characterize the T-gate density $z$ that is necessary to generate typical states using Clifford+T circuits. We find that the curves of $C_M$ and $\partial_z C_M/N$ for different $N$ intersect at a single point at $z_{c,C_M} \approx 2.35$, indicating a saturation transition at $z_{c,C_M}$. The transition is accompanied by a shift in the scaling of $C_M$, from volume-law to being independent of $N$. This saturation transition also places a lower bound of $z \gtrapprox 2.35$ needed for Clifford+T circuits to approximate Haar random states [34, 75, 76]. This contrasts 8-point OTOCs or $M_1$, which are less accurate indicators of Haar randomness, as they saturate already for smaller $z \gtrapprox 2$ [76]. The saturation transition in $C_M$ also implies a sharp change in the number of Pauli samples $K$ needed to estimate $M_1$: The estimation of $M_1$ requires only a number of samples which is independent of $N$ whenever $z \gtrapprox z_{c,C_M}$.

In fact, we find that the magic capacity beyond the transition point is consistent with that of typical states. In Ref. [79], a quantity called filtered SRE was introduced to distinguish the magic of typical and atypical states. It was shown that the density of the filtered SRE, as a function of the Rényi index $\alpha$, exhibits distinct behavior for these two classes of states. The filtered SRE was specifically designed to address the problem that this distinguishing feature is exponentially suppressed in the standard SRE. Our work establishes that this distinction can also be observed in the magic capacity, offering an alternative method for distinguishing typical and atypical states, which we corroborate through our numerical results. Because the distinguishing feature lies in the *scaling* of magic capacity, rather than simply the density as in the filtered SRE, this new method offers a more pronounced separation between typical and atypical states. Furthermore, it has a particular advantage of efficient computability via Pauli sampling.

Finally, we investigate the magic capacity in the ground state of the Heisenberg and TFIM. For the Heisenberg model, we find that for $\Delta \ll 1$, the magic capacity scales linearly with system size, $C_M \propto N$. However, at $\Delta \approx 1$, the scaling transitions to $C_M \propto N^2$, which asymptotically saturates the known upper bound $C_M = O(N^2)$. For the Ising model, we find that the capacity scales as $C_M \propto N$ for all $h$, but exhibits a characteristic dip at the critical point $h = 1$. This highlights that the magic capacity can be used to characterize the ground states of Hamiltonians.

Our work opens several avenues for future research. First, magic capacity could be applied as an efficient tool to characterize transitions of magic in random quantum circuits, driven by competition of Clifford and non-Clifford resources [42–44, 80]. Furthermore, it is instructive to connect the mutual $\alpha$-SRE to the notion of long-range magic as magic that cannot be removed by finite-depth circuits [57]. Finally, it would be worth exploring other aspects of magic that may offer enhanced robustness in detecting quantum correlations, criticality, and other crucial physical phenomena, such as topological order or gauge theory. This could uncover novel connections between magic and fundamental quantum properties.

## Acknowledgments

We thank Lorenzo Piroli, David Aram Korbany, Marcello Dalmonte, and Emanuele Tirrito for insightful discussions. Our MPS simulations have been performed using the iTensor library [81].

**Funding information** P.S.T. acknowledges funding by the Deutsche Forschungsgemeinschaft (DFG, German Research Foundation) under Germany's Excellence Strategy – EXC-2111 – 390814868.

We provide proofs and additional details supporting the claims in the main text.

# A  Proof of marginals for probability distribution $q_\rho(P)$

In the main text, we consider the alternative probability distribution over Pauli strings $P$ as

$$q_\rho(P) = 2^{-N}\mathrm{tr}(\rho P \rho P). \tag{A.1}$$

A nice property of $q_\rho$ is that, for any subsystem $A$ and its complement $A^c$, the distribution for the reduced density matrix $\rho_A = \mathrm{tr}_{A^c}(\rho)$ is simply the marginal distribution of $q_\rho$ in $A$. In other words,

$$q_{\rho_A}(P_A) = \sum_{P \in \mathcal{P}_{A^c}} q_\rho(P_A \otimes P), \tag{A.2}$$

for any $P_A \in \mathcal{P}_A$.

This can be shown as follows. First, we rewrite

$$\sum_{P \in \mathcal{P}_{A^c}} q_\rho(P_A \otimes P) = 2^{-N}\mathrm{tr}\left(P_A \otimes I_{A^c} \rho P_A \otimes I_{A^c} \sum_{P \in \mathcal{P}_{A^c}} I_A \otimes P \rho I_A \otimes P\right). \tag{A.3}$$

Now, we can see that $\sum_{P \in \mathcal{P}_{A^c}} I_A \otimes P \rho I_A \otimes P$ is a Pauli twirl over partition $A^c$, which is a 1-design and thus leaves only the identity invariant. In particular, one can repeatedly use the identity $\sum_{P \in \mathcal{P}_1} P(.)P = 2I_1\mathrm{tr}_1(.)$ [17] to show that

$$\sum_{P \in \mathcal{P}_{A^c}} I_A \otimes P \rho I_A \otimes P = 2^{N_{A^c}} \rho_A \otimes I_{A^c}. \tag{A.4}$$

Using this result, we get

$$\begin{aligned}
\sum_{P \in \mathcal{P}_{A^c}} q_\rho(P_A \otimes P) &= 2^{-N_A}\mathrm{tr}(P_A \otimes I_{A^c} \rho P_A \otimes I_{A^c} \rho_A \otimes I_{A^c}) \\
&= 2^{-N_A}\mathrm{tr}(\mathrm{tr}_{A^c}(P_A \otimes I_{A^c} \rho P_A \otimes I_{A^c} \rho_A \otimes I_{A^c}) \\
&= 2^{-N_A}\mathrm{tr}(P_A \rho_A P_A \rho_A)) \equiv q_{\rho_A}(P_A).
\end{aligned} \tag{A.5}$$

# B  Comparison of mutual SREs

In this section, we compare the different types of mutual SREs as introduced in the main text. We study the mutual 2-SRE $\mathcal{I}_2$, mutual von-Neumann SRE $\mathcal{I}_1^{[p]}$ via $\mathrm{tr}(\rho P)^2$, an $\mathcal{I}_1^{[q]}$ via $\mathrm{tr}(\rho P \rho P)$.

First, we study the ground state of the TFIM in Fig. 7a where we choose a low number of qubits $N = 12$ and large number of qubits $N = 80$ in Fig. 7b. For $N = 80$, we estimate $\mathcal{I}_1^{[p]}$ using $K = 2 \times 10^6$ Pauli samples, while for $\mathcal{I}_1^{[q]}$ we use much lower $K = 10^5$ samples. We note that $\mathcal{I}_1^{[p]}$ requires significantly more samples to estimate due the $p$ distribution of a subsystem not being a marginal for the full distribution, thus increasing the variance of the estimator.

Next, we study the Heisenberg model in Fig. 8. We find find that $\mathcal{I}_2$ decreases towards $\Delta = 1$, while $\mathcal{I}_1^{[q]}$ and $\mathcal{I}_1^{[p]}$ increase. Notably, we find that for large $N$ and $\Delta \approx 1$, $\mathcal{I}_1^{[q]}$ and $\mathcal{I}_1^{[p]}$ show different scalings: $\mathcal{I}_1^{[q]}$ is constant, while $\mathcal{I}_1^{[p]}$ scales with $N$.

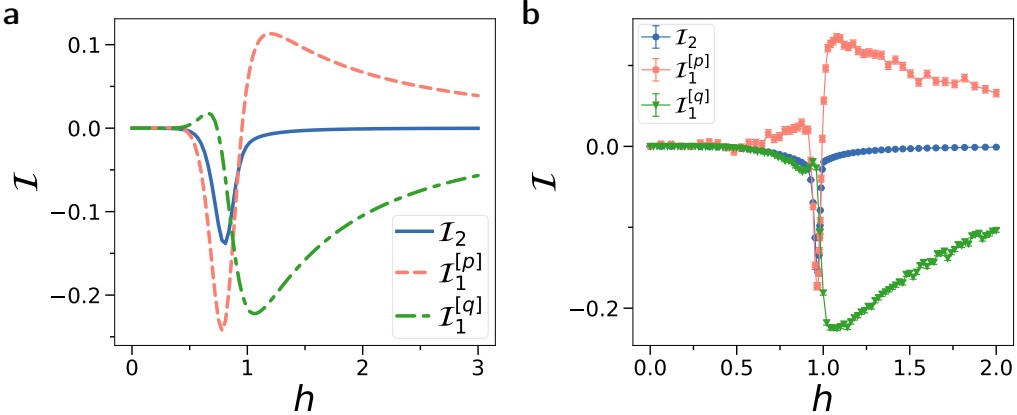

Figure 7: Comparison of mutual 2-SRE $\mathcal{I}_2$, mutual von-Neumann SRE $\mathcal{I}_1^{[p]}$ via $\mathrm{tr}(\rho P)^2$, and $\mathcal{I}_1^{[q]}$ via $\mathrm{tr}(\rho P \rho P)$. We show bipartition at the center of the chain with **a)** $N = 12$ and **b)** $N = 80$ qubits in total.

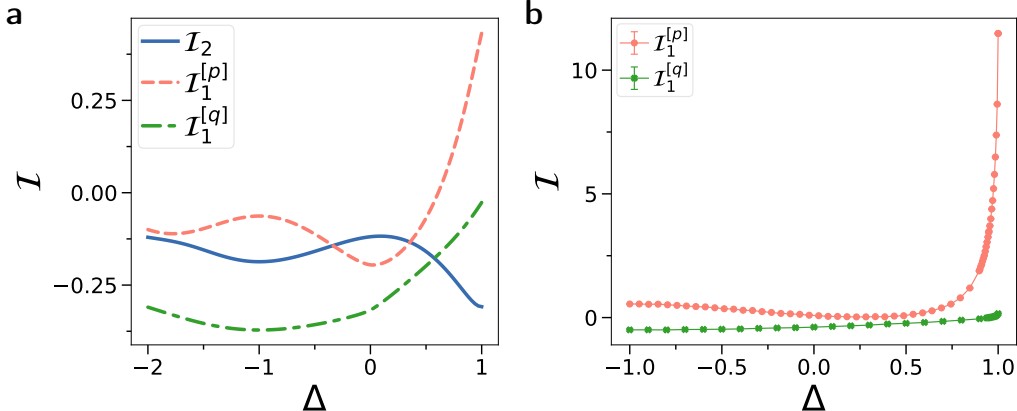

Figure 8: Different types of mutual magic for ground state of Heisenberg model for anisotropy $\Delta$. Comparison of mutual 2-SRE $\mathcal{I}_2$, mutual von-Neumann SRE $\mathcal{I}_1^{[q]}$ via $\mathrm{tr}(\rho P \rho P)$, and its formulation $\mathcal{I}_1^{[p]}$ via $\mathrm{tr}(\rho P)^2$. We show bipartition at the center of the chain with **a)** $N = 12$ and **b)** $N = 80$ qubits in total.

Now, we study the scaling of mutual magic $\mathcal{I}$ in more detail. While mutual magic is usually independent of $N$, we find classes of states where it scales with $N$. In particular, we study a magical GHZ state

$$|\psi_{\mathrm{mGHZ}}\rangle = \frac{1}{\sqrt{2}} \exp(-i\pi/8\sigma^z)^{\otimes N} \exp(-i\beta/2\sigma^y)^{\otimes N} (|00\ldots0\rangle + |11\ldots1\rangle), \tag{B.1}$$

where each qubit is rotated by $\exp(-i\pi/8\sigma^z)\exp(-i\beta/2\sigma^y)$ with $\beta = \arccos(1/\sqrt{3})$, which corresponds to the single-qubit rotation that induces the most magic. For $\mathcal{I}_1^{[q]}(|\psi_{\mathrm{mGHZ}}\rangle)$ and $\mathcal{I}_2(|\psi_{\mathrm{mGHZ}}\rangle)$, the mutual magic is positive and constant with $N$. In contrast, we observe $\mathcal{I}_1^{[p]}(|\psi_{\mathrm{mGHZ}}\rangle) \propto -N$. We believe this may originate from the fact that the $p$ distribution on subsystems is not a marginal distribution with respect to the full system. This finding indicates that the mutual magic based on $p$ distribution may not be related to the notion of long-range magic as magic that cannot be removed by finite-depth circuits [57,58], and future work may focus on $q$ distribution to establish a connection.

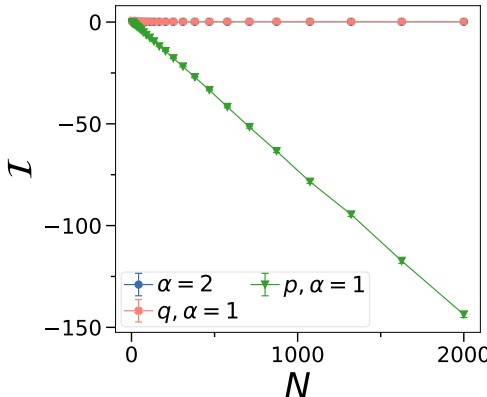

Figure 9: Different types of mutual magic for a magical GHZ state as defined in Eq. (B.1). Comparison of mutual 2-SRE $\mathcal{I}_2$, mutual von-Neumann SRE $\mathcal{I}_1^{[q]}$ via $\mathrm{tr}(\rho P \rho P)$, and its formulation $\mathcal{I}_1^{[p]}$ via $\mathrm{tr}(\rho P)^2$. We show bipartition at the center of the chain with $N = 80$ qubits in total.

## C  Magic capacity of typical states

The phenomenological SRE of typical states was derived in Ref. [79], which reads for large $N$

$$M_\alpha^{\mathrm{typ}} = \frac{1}{1-\alpha} \ln \left[ \frac{(\eta-1)(2b)^\alpha \Gamma\left(\alpha + \frac{1}{2}\right)}{\sqrt{\pi} d} + \frac{1}{d} \right], \tag{C.1}$$

where $\Gamma(x)$ is the gamma function, $b = (d/2+1)^{-1}$, $d = 2^N$, and $\eta = d^2$ ($\eta = d(d+1)/2$) for complex (real) typical states. In the limit of large $N$, the average von-Neumann SRE is given by

$$M_1^{\mathrm{typ}} = \lim_{\alpha \to 1} M_\alpha^{\mathrm{typ}} = N \ln(2) - 2\ln(2) - \frac{\Gamma'(3/2)}{\Gamma(3/2)}, \tag{C.2}$$

and the magic capacity, computed using Eq. (16), is given by

$$C_M^{\mathrm{typ}} = \frac{\Gamma''(3/2)}{\Gamma(3/2)} - \left( \frac{\Gamma'(3/2)}{\Gamma(3/2)} \right)^2 \approx 0.934802\ldots \tag{C.3}$$

We see that the magic capacity becomes constant in the limit $N \to \infty$.

This behavior can be contrasted with the magic capacity of the (atypical) tensor product of single-qubit state $|\Theta\rangle = \cos(\theta/2)|0\rangle + e^{-i\phi}\sin(\theta/2)|1\rangle$ for generic $0 < \theta < \pi/2$ and $\phi$. We have that

$$C_M\left(|\Theta\rangle^{\otimes N}\right) = N C_M(|\Theta\rangle), \tag{C.4}$$

i.e. the magic capacity scales linearly with system size. This distinct scaling behavior clearly distinguishes typical and atypical states. Note that $C_M$ is bounded as $C_M = O(N^2)$, where we find that the ground state of the Heisenberg model for $\Delta = 1$ satisfies such scaling.

## D  Mutual von-Neumann SRE for MPS

We now show that the mutual von-Neumann SRE $\mathcal{I}_1^{[q]}(|\psi\rangle)$ can be efficiently computed with time $O(N\chi^3\epsilon^{-2})$ for MPS

$$|\psi\rangle = \sum_{\{s_k\}} A_1^{s_1} \ldots A_N^{s_N} |s_1, \ldots, s_N\rangle, \tag{D.1}$$

where $A_k^s$ are matrices with maximal size $\chi \times \chi$. We assume two complementary bipartitions $A$ and $B$. The 2-Rényi entanglement $S_2(\rho_A)$ and $S_2(\rho_B)$ can be efficiently computed for MPS via standard methods, where $\rho_A$, $\rho_B$ are the reduced density matrix of the two bipartitions. Then, $\mathcal{I}_1^{[q]}(|\psi\rangle)$ can be computed as

$$\mathcal{I}_1^{[q]}(\rho) = \mathop{\mathbb{E}}_{P \sim q(P)} [\ln(\text{tr}(\rho P \rho P)) - \ln(\text{tr}(\rho_A P_A \rho_A P_A)) - \ln(\text{tr}(\rho_B P_B \rho_B P_B))] + S_2(\rho_A) + S_2(\rho_B),$$

where $P_A$ and $P_B$ are Pauli string $P$ reduced onto subsystem $A$ and $B$ respectively, such that $P = P_A \otimes P_B$. Crucially, we need to sample from $q(P)$ which can be done in $O(N\chi^3)$ as shown in Ref. [17]. In particular, this can be done by ancestral sampling, using the following relation:

$$q(P) = q(P_1)q(P_2|P_1)\ldots q(P_N|P_1 \otimes P_2 \otimes \cdots \otimes P_N), \tag{D.2}$$

where $P_i \in \{I, \sigma^x, \sigma^y, \sigma^z\}$ are single-qubit Pauli operators acting on the $i$th qubit and $q(P_2|P_1)$ conditional probabilities.

We start by computing the probabilities for the reduced density matrix $\rho_1 = \text{tr}_{2,\ldots N}(\rho)$ over the first qubit

$$q(P_1) = \sum_{P_2,\ldots,P_N} q(P_1, P_2, \ldots, P_N) = \frac{1}{2}\text{tr}(\rho_1 P_1 \rho_1 P_1), \tag{D.3}$$

for all $P_1$. Here, we used the fact that after tracing out the $k$th qubit of $\rho$, the probability distribution over the reduced system can be written as

$$q_{\text{tr}_k(\rho)}(P) = \sum_{m=0}^{3} q_\rho(P \otimes P_k^m). \tag{D.4}$$

We sample a Pauli operator $P_1^*$ for the first qubit according to $q(P_1)$. Then, we proceed with the second qubit as

$$q(P_2|P_1^*) = \sum_{P_3,\ldots,P_N} q(P_1^*, P_2, \ldots, P_N) = \frac{2^{-2}\text{tr}(\rho_{1,2} P_1^* \otimes P_2 \rho_{1,2} P_1^* \otimes P_2)}{2^{-1}\text{tr}(\rho_1 P_1^* \rho_1 P_1^*)},$$

where $\rho_{1,2}$ is the reduced density matrix of $\rho$ over the first two qubits. Then, we sample a $P_2$ from $q(P_2|P_1^*)$. These steps are now repeated until $N$ qubits are reached, finally gaining a Pauli $P \sim q_\rho(P)$. For improved efficiency, the MPS can first be brought to the left canonical form before performing the sampling procedure, as discussed in detail in Ref. [18]. After sampling $P$, one has to evaluate terms of the form $\text{tr}(\rho P \rho P)$ and $\text{tr}(\rho_A P_A \rho_A P_A)$, which can be done in $O(N\chi^3)$. In particular, the contractions for terms

$$\text{tr}\left(\rho_A P_1 \otimes P_2 \otimes \cdots \otimes P_{N_A} \rho_A P_1 \otimes P_2 \otimes \cdots \otimes P_{N_A}\right), \tag{D.5}$$

with $\rho_A = \text{tr}_{\bar{A}}(\rho)$ can be done as shown in Fig. 10 with time complexity $O(N\chi^3)$ for MPS of bond dimension $\chi$. Here, $A_i$ denotes the tensor corresponding to the $i$th qubit of the MPS, $P_i$ are the Pauli operator for the $i$th qubit, and lines are contracted over. For optimal complexity, one first contracts over the complement of subspace $A$, then the bonds that involve Pauli operators, and finally the remaining bonds.

# E   Statevector simulation of Pauli sampling

We now show an improved statevector algorithm for Pauli sampling and Bell sampling. Here, by statevector simulation we mean that we describe the state by its full statevector amplitude $|\psi\rangle = \sum_i a_i |i\rangle$ which describes arbitrary quantum states. In contrast to naive sampling,

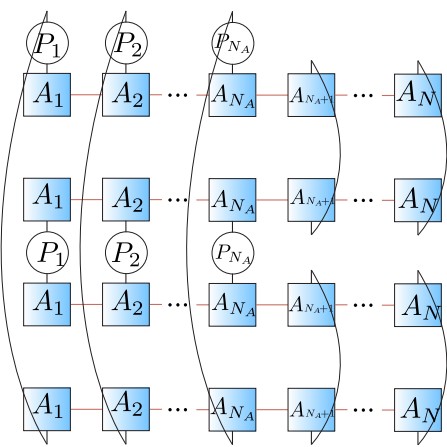

Figure 10: MPS contraction to compute Eq. (D.5).

which takes $O(8^N)$ time and $O(4^N)$ memory, our method requires only $O(8^{N/2})$ time and $O(2^N)$ memory. While MPS Pauli sampling in $O(N\chi^3)$ has the same asymptotic scaling (for worst case $\chi = 2^{N/2}$) [17,18], we note that our algorithm directly works in the statevector representation of states, which is beneficial when working with states already given in statevector representation.

Note that our algorithm perfectly samples from the Pauli distribution directly, and does not use Monte-Carlo approaches to approximately sample. We apply our method to compute the von-Neumann SRE and magic capacity in Sec. F.

First, we define the Pauli matrices $\sigma_{00} = I_2$, $\sigma_{01} = \sigma^x$, $\sigma_{10} = \sigma^z$ and $\sigma_{11} = \sigma^y$. The $4^N$ Pauli strings are $N$-qubit tensor products of Pauli matrices which we define as $\sigma_{\boldsymbol{n}} = \bigotimes_{j=1}^N \sigma_{n_{2j-1}n_{2j}}$ with $\boldsymbol{n} \in \{0,1\}^{2N}$. The Bell states are given by $|\sigma_{00}\rangle = \frac{1}{\sqrt{2}}(|00\rangle + |11\rangle)$, $|\sigma_{01}\rangle = \frac{1}{\sqrt{2}}(|00\rangle - |11\rangle)$, $|\sigma_{10}\rangle = \frac{1}{\sqrt{2}}(|01\rangle + |10\rangle)$ and $|\sigma_{11}\rangle = \frac{1}{\sqrt{2}}(|01\rangle - |10\rangle)$ and we define the product of Bell states $|\sigma_{\boldsymbol{r}}\rangle = |\sigma_{r_1 r_2}\rangle \otimes \cdots \otimes |\sigma_{r_{2N-1}r_{2N}}\rangle$.

We now want to sample from the probability distribution $p_{|\psi\rangle}(\sigma) = 2^{-N}\langle\psi|\sigma|\psi\rangle^2$. This can be done by sampling the Bell basis via [64,69]

$$p(\sigma_{\boldsymbol{r}}) = 2^{-N}|\langle\psi|\sigma_{\boldsymbol{r}}|\psi\rangle|^2 = |\langle\boldsymbol{r}|\eta\rangle|^2, \tag{E.1}$$

where $|\boldsymbol{r}\rangle$ is the computational basis state corresponding to bitstring $\boldsymbol{r}$. Here, we have the Bell transformed state $|\eta\rangle = U_{\text{Bell}}^{\otimes N}|\psi^*\rangle \otimes |\psi\rangle$ with Bell transformation $U_{\text{Bell}} = (H \otimes I_1)\text{CNOT}$, where $H = \frac{1}{\sqrt{2}}(\sigma^x + \sigma^z)$ is the Hadamard gate, $\text{CNOT} = \exp\left(i\frac{\pi}{4}(I_1 - \sigma^z) \otimes (I_1 - \sigma^x)\right)$ and $|\psi^*\rangle$ is the complex conjugate of $|\psi\rangle$. Thus, sampling $|\boldsymbol{r}\rangle$ in the computational basis from $|\eta\rangle$ is equivalent to sampling $\sigma_{\boldsymbol{r}}$ from $p(\sigma_{\boldsymbol{r}})$. However, storing the $2N$-qubit state $|\eta\rangle$ with the Hilbert space dimension of $4^N$ in memory is a major bottleneck.

To solve this problem, we provide a hybrid Schrödinger-Feynmann algorithm to sample $\sigma_{\boldsymbol{r}} \sim p(\sigma_{\boldsymbol{r}})$ which combines Bell and Feynmann-type approach for sampling for improved performance:

**Theorem E.1** (Simulation of Pauli sampling in statevector representation)**.** *Given an $N$-qubit state $|\psi\rangle = \sum_{i=1}^{2^N} a_i |i\rangle$, there is an algorithm to sample $\sigma$ from the probability distribution $p(\sigma) = 2^{-N}|\langle\psi|\sigma|\psi\rangle|^2$ in $O(8^{N/2})$ time and $O(2^N)$ memory within the statevector representation of $|\psi\rangle$.*

*Proof.* The overall idea is to sample the first $N$ qubits from $|\eta\rangle$ in a Feynmann-like algorithm, then switch to direct sampling with marginals for the last $N$ qubits.

We start with the Feynmann-like part of our algorithm. Any $N$-qubit state $|\phi\rangle$ can be written as Schmidt-decomposition with $|\phi\rangle = a_0 |0\rangle |\phi_0\rangle + a_1 |1\rangle |\phi_1\rangle$, where $|0\rangle, |1\rangle$ are computational basis states and $|\phi_0\rangle, |\phi_1\rangle$ are normalized $N-1$ qubit states. For two copies, we have

$$|\phi^*\rangle |\phi\rangle = \sum_{k,q=1}^{2} a_k^* a_q |k\rangle \big|\phi_k^*\big\rangle |q\rangle \big|\phi_q\big\rangle . \tag{E.2}$$

We now reorder the positions of the qubits such that the first qubit of each copy are placed first. Then, we apply the Bell transformation $U_{\text{Bell}}$ on the first qubit of each copy

$$
\begin{aligned}
\big|\eta'\big\rangle &= U_{\text{Bell}} \otimes I^{\otimes 2N-2} \sum_{k,q=1}^{2} a_k^* a_q |k\rangle |q\rangle \big|\phi_k^*\big\rangle \big|\phi_q\big\rangle \\
&= \frac{1}{\sqrt{2}} \Big[ |00\rangle \big(|a_0|^2 \big|\phi_0^*\big\rangle |\phi_0\rangle + |a_1|^2 \big|\phi_1^*\big\rangle |\phi_1\rangle\big) \\
&\qquad + |01\rangle \big(a_0^* a_1 \big|\phi_0^*\big\rangle |\phi_1\rangle + a_1^* a_0 \big|\phi_1^*\big\rangle |\phi_0\rangle\big) \\
&\qquad + |10\rangle \big(|a_0|^2 \big|\phi_0^*\big\rangle |\phi_0\rangle - |a_1|^2 \big|\phi_1^*\big\rangle |\phi_1\rangle\big) \\
&\qquad + |11\rangle \big(a_0^* a_1 \big|\phi_0^*\big\rangle |\phi_1\rangle - a_1^* a_0 \big|\phi_1^*\big\rangle |\phi_0\rangle\big) \Big] .
\end{aligned}
\tag{E.3}
$$

We then measure the first two qubits $|nm\rangle$ in the computational basis with the probability

$$P(|nm\rangle) = \big\langle \eta'\big| (|nm\rangle \langle nm| \otimes I^{\otimes 2N-2}) \big|\eta'\big\rangle .$$

As the tensor state $\big|\phi_n^*\big\rangle |\phi_m\rangle$ is too large to be stored directly in memory, we instead compute the sampling probabilities via the overlaps between the states $|\phi_n\rangle$ with the overlap matrix $E_{nm} = a_n^* a_m \langle \phi_n | \phi_m \rangle$. For example for outcome $|00\rangle$, we have

$$P(|00\rangle) = \frac{1}{2} \left( |E_{00}|^2 + E_{01}^* E_{10} + E_{10}^* E_{01} + |E_{11}|^2 \right) ,$$

and the normalized projected state

$$|\eta_{00}\rangle = \frac{1}{\sqrt{2 P(|0\rangle |0\rangle)}} \left( |a_0|^2 \big|\phi_0^*\big\rangle |\phi_0\rangle + |a_1|^2 \big|\phi_1^*\big\rangle |\phi_1\rangle \right) . \tag{E.4}$$

We now compute $P(|nm\rangle)$ and sample from it. After sampling, we gain two bits $n$, $m$ which indicate the first Pauli operator $\sigma_{nm}$ of the full Pauli string, as well as the normalized projected state $|\eta_{nm}\rangle$. Note that we do not store $|\eta_{nm}\rangle$ in memory directly, but only its amplitudes $a_0$, $a_1$ and the states $|\phi_0\rangle, |\phi_1\rangle$.

Starting with the projected state $|\eta_{nm}\rangle$, we can now repeat above steps to sample more qubits. For example, continuing with the example $|\eta_{00}\rangle$ of Eq. (E.4), this state consists of two superposition states $\big|\phi_0^*\big\rangle |\phi_0\rangle$ and $\big|\phi_1^*\big\rangle |\phi_1\rangle$. On each superposition state, we repeat the step of Eq. (E.3) and sample the next two qubits, gaining another two bits $k\ell$. Now, the projected state of $2N-4$ qubits $|\eta_{nmk\ell}\rangle$ is described by 4 superposition states. After $k$ repetitions, we have sampled $2k$ bits $\boldsymbol{r}'$ which correspond to the Pauli operators for the first $k$ qubits. The projected state $|\eta_{\boldsymbol{r}'}\rangle$ has $2N-2k$ qubits and is described by a superposition of $2^k$ states. After $k = N$ steps, we would have sampled the full Pauli string. The number of superposition state needed scales exponentially with $k$. Here, the main complexity arises from computing the overlap matrix $E$ needed for sampling, which has $(2^k + 1)2^k/2$ non-trivial entries. The time complexity scales as $O(4^k 2^{N-k})$, while the memory consumption is $O(4^k)$.

If we were to continue sampling until the last qubits, there would be no advantage in terms of computational effort. However, we can reduce complexity by stopping early and doing only

$k = N/2$ sampling steps, which has a complexity of $O(8^{N/2})$. At this point, the projected state of $N$ qubits $|\eta_{r'}\rangle$ is a superposition state of $2^{N/2}$ states. Note that the $2^{N/2}$ states that make up the superposition state take only $O(2^N)$ memory to store in total. After $k = N/2$ sampling steps, we switch to a direct sampling approach by explicitly constructing the state $|\eta_{r'}\rangle$ of dimension $2^N$ by summing over the $2^{N/2}$ linear combinations of states with their corresponding coefficients. Constructing the state explicitly has a time complexity of $O(8^{N/2})$ and memory complexity $O(2^N)$. Then, we directly sample from $|\eta_r\rangle$ via its marginals in a time $O(2^N)$. This completes sampling of a Pauli string $\sigma_r$ according to the probability distribution $p(\sigma_r)$. Our Bell sampling simulation runs in a time of $O(8^{N/2})$ and requires $O(2^N)$ memory. $\qquad\square$

In contrast to Pauli sampling (which is inefficient on quantum computers [82]) the closely related Bell sampling can be done efficiently in experiment. It samples from distribution $2^{-N}\langle\psi^*|\sigma|\psi\rangle^2$ [64] which can be sampled from via $U_{\text{Bell}}^{\otimes N}|\psi\rangle\otimes|\psi\rangle$. Our algorithm can also be used to perform Bell sampling. Here, one uses a modified Pauli sampling algorithm where one replaces the complex conjugate $|\psi^*\rangle$ at every step of the algorithm with the non-conjugated state $|\psi\rangle$.

## F Improved statevector computation of von-Neumann SRE and magic capacity

Now, we apply our Pauli sampling algorithm for statevector simulation to compute SREs and magic capacity. First, we want to compute the von-Neumann SRE

$$M_1(|\psi\rangle) = -\sum_{\sigma\in\mathcal{P}} p(\sigma)\ln\big(\langle\psi|\sigma|\psi\rangle^2\big) = -\mathop{\mathbb{E}}_{\sigma\sim p(\sigma)}\big[\ln\big(\langle\psi|\sigma|\psi\rangle^2\big)\big]. \tag{F.1}$$

We perform Pauli sampling $K$ times, gaining $K$ Pauli strings $\{r_j\}_{j=1}^K$. From this, we compute the unbiased estimator of $M_1$

$$\hat{M}_1 = -\frac{1}{K}\sum_{j=1}^K \ln\big(\langle\psi|\sigma_{r_j}|\psi\rangle^2\big), \tag{F.2}$$

where $\langle\psi|\sigma_{r_j}|\psi\rangle^2$ can be evaluated in $O(2^N)$ time. The estimator has a variance

$$C_M = \text{var}(\hat{M}_1) = \mathop{\mathbb{E}}_{P\sim p(P)}\Big[\ln\big(\langle\psi|P|\psi\rangle^2\big)^2\Big] - M_1^2, \tag{F.3}$$

which is given by the magic capacity $C_M$. When averaging over $K$ samples, the estimated mean value $\bar{M}_1$ has a standard deviation $\sqrt{C_M/K}$.

Now, we want to estimate the simulation complexity to achieve a given precision $\epsilon$ in the estimation of $M_1$. For this we need to bound the variance of our estimator given by $C_M = \text{var}(\hat{M}_1)$. One can upper bound this via $\text{var}(\ln\big(\langle\psi|\sigma|\psi\rangle^2\big)) \le N^2\ln(2)^2 + 1$ [18]. Thus, the estimation error $\epsilon$ of the average $\bar{M}_1$ scales in the worst case as

$$\epsilon \sim \sqrt{C_M/K} \sim \frac{N}{\sqrt{K}}. \tag{F.4}$$

Note that to estimate the SRE density $m_1 = M_1/N$, we have similarly

$$\tilde{\epsilon} \propto \sqrt{C_M/(KN^2)} \sim \frac{1}{\sqrt{K}}. \tag{F.5}$$

Overall, our algorithm scales as $O(8^{N/2}\epsilon^{-2})$ in time and $O(2^N)$ in memory, giving a square-root speedup over naive methods to compute SREs for statevectors. In particular, our method can compute the von-Neumann SRE of arbitrary states for 24 qubits using a standard notebook, while naive methods are limited to 15 qubits.

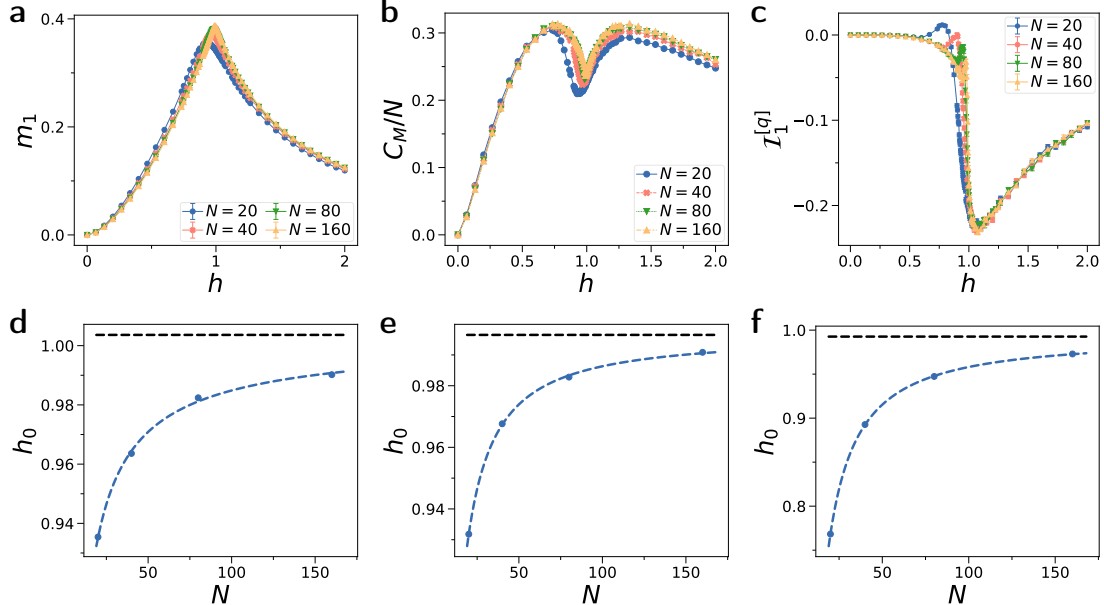

Figure 11: Determining critical point $h_c$ for groundstate of TFIM against field $h$, where we choose the Hamiltonian in the standard computational basis. We track the extremal value and the corresponding field $h_0(N)$ for different $N$. Then we fit $h_0(N)$ with polynomial $h_0(N) = -cN^{(-\gamma)} + h_c^{\text{fit}}$ to determine $h_c^{\text{fit}}$. **a)** $m_1 = M_1/N$, **b)** $C_M/N$ and **c)** mutual von-Neumann SRE $\mathcal{I}_1^{[q]}$. Next, we plot $h_0(N)$ against $N$. We fit $h_0(N)$ with $h_0(N) = -cN^{-\gamma} + h_c^{\text{fit}}$, where corresponding fit $h_c^{\text{fit}}$ as horizontal dashed line. We have **d)** $m_1$ with $h_c^{\text{fit}} \approx 1.0036$, **e)** $C_M$ with $h_c^{\text{fit}} \approx 0.9965$, and **f)** $\mathcal{I}_1^{[q]}$ with $h_c^{\text{fit}} \approx 0.9926$.

# G   Critical point of TFIM via SREs and magic capacity

Here, we show how to determine the critical point from the SREs and magic capacity. The TFIM has a critical point at $h = 1$, where the ground state acquires universal behavior.

It has been shown that in the standard basis, the SRE $M_2$ can be used to determine the critical point [16]. Here, we use $M_1$, $\mathcal{I}_1$ and $C_M$ to determine the critical point. In Fig. 11, we plot $m_1$, $C_M/N$ and $\mathcal{I}_1^{[q]}$ against $h$ for different $N$. Close to $h = 1$, we find that $m_1$ and $\mathcal{I}_1^{[q]}$ have a sharp peak, while $C_M/N$ shows a minima. We record the extremal value and corresponding field $h_0$ for all $N$. Then, we fit $h_0(N)$ with $h_0(N) = -cN^{-\gamma} + h_c^{\text{fit}}$ and use $h_c^{\text{fit}}$ as approximation of the critical point. We find that $m_1$, $C_M/N$ and $\mathcal{I}_1^{[q]}$ provide accurate estimations of the critical points.

# H   Mutual SRE for basis-independent critical point of TFIM

A question raised by Ref. [16] was how to estimate the critical point $h_c$ of the TFIM ground state via magic. It has been noted that in the standard basis, the magic of the ground state shows a clear universal peak at the critical point $h_c = 1$. However, this is not true when one rotates the Hamiltonian into a different local basis via single-qubit unitaries $V$ over all qubits $N$, i.e. $H' = V^{\otimes N} H V^{\otimes N\dagger}$. Then, in general, the magic of the ground state of the rotated Hamiltonian does not have a clear extremum at the critical point [16]. The critical point can be identified only by applying additional fitting procedures.

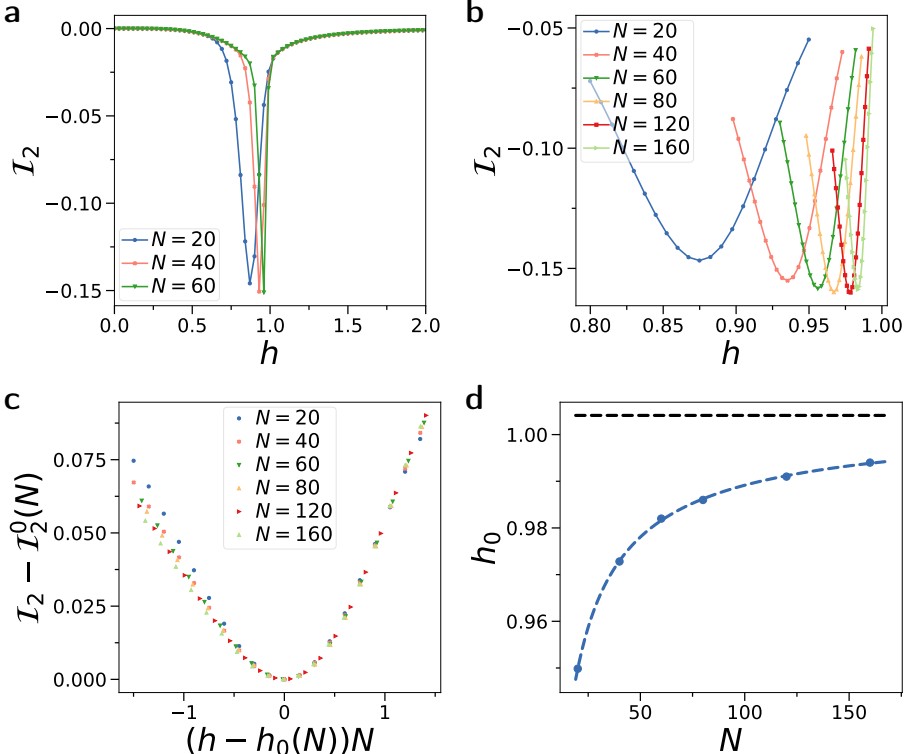

Figure 12: Mutual 2-SRE $\mathcal{I}_2$ for equal bipartition of size $N/2$ close to criticality of the TFIM. **a)** $\mathcal{I}_2$ plotted against field $h$. **b)** $\mathcal{I}_2$ plotted against field $h$ around critical point $h = 1$. **c)** $\mathcal{I}_2 - \mathcal{I}_2^0$ plotted against shifted field $h - h_0$ rescaled with $N$. We define $\mathcal{I}_2^0(N) = \max_h \mathcal{I}$ as the maximal mutual SRE over field $h$, which becomes maximal for $h_0(N)$. **d)** We plot $h_0(N)$ against $N$. We fit $h_0(N)$ with $h_0(N) = -cN^{-\gamma} + h_c^{\text{fit}}$, where $h_c^{\text{fit}} \approx 1.004$ is the fitted critical $h_c$ for $N \to \infty$, which we show as horizontal dashed line.

We now argue that the mutual SRE is a good indicator to find the critical point independent of the local basis.

First, in Fig.12 we consider the standard local basis with the mutual 2-SRE $\mathcal{I}_2$ of the ground state of the TFIM as function of $h$. In Fig.12a, we see that the mutual SRE for different $N$ has a clear peak close to the critical point $h = 1$, where for increasing $N$ the peak is approaching $h = 1$ asymptotically. Note that the peak is far more pronounced compared to just regarding SRE $m_2$. We see the convergence of the extremum to $h = 1$ with $N$ more closely in Fig.12b. In Fig.12c, we plot $\mathcal{I}_2 - \mathcal{I}_2^0$, where subtract the maximal mutual SRE $\mathcal{I}_2^0$, against $(h - h_0)N$, where $h_0(N)$ is the field where we find the maximal mutual SRE for different $N$. We find that the rescaled curves collapse to a single curve close to the critical point. In Fig.12d, we plot the minimal $h_0(N)$ against $N$. We fit with $h_0(N) = -cN^{(-\gamma)} + h_c^{\text{fit}}$, where $h_0(N \to \infty) = h_c^{\text{fit}}$ is the fitted critical field, which we show as dashed horizontal line. We find from the fit $h_c^{\text{fit}} \approx 1.004$, which is close to the analytic value $h_c = 1$.

Next, we study study the critical point for different local basis. We rotate $H' = V^{\otimes N} H V^{\otimes N\dagger}$ with local unitary $V$ we choose from $V_\alpha(\theta) = \exp(-i\sigma^\alpha \theta/2)$, where $\sigma^\alpha$ is a Pauli matrix.

First, we study $V_z(\theta) = \exp(-i\sigma^z \theta/2)$ with $\theta = \pi/4$. We show $m_2$ in Fig. 13a, finding the absence of an extremum at $h = 1$. In contrast, we find in Fig. 13b that $\mathcal{I}_2$ is clearly peaked towards $h = 1$, where we find a minimum that converges towards $h = 1$ large $N$.

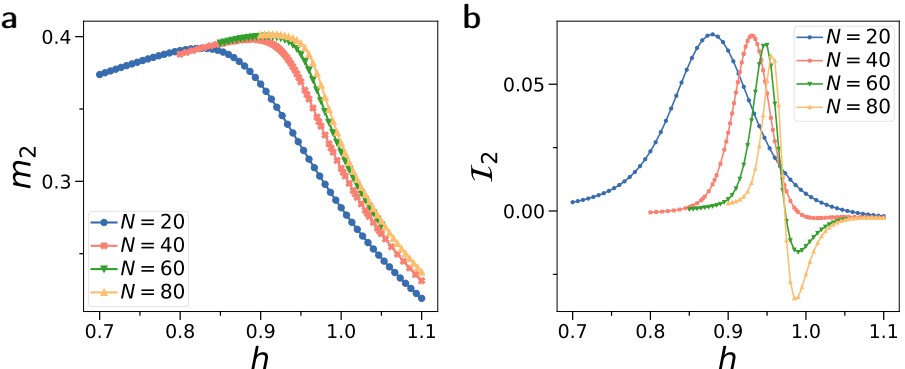

Figure 13: Magic density $m_2$ and mutual 2-SRE $\mathcal{I}_2$ close to criticality of the TFIM in local basis rotated with single-qubit unitary $V_z(\theta) = \exp(-i\sigma^z\theta/2)$ with $\theta = \pi/4$. **a)** $m_2$ against $h$ for $\chi = 10$. **b)** $\mathcal{I}_2$ with equal bipartition $N/2$ against $h$.

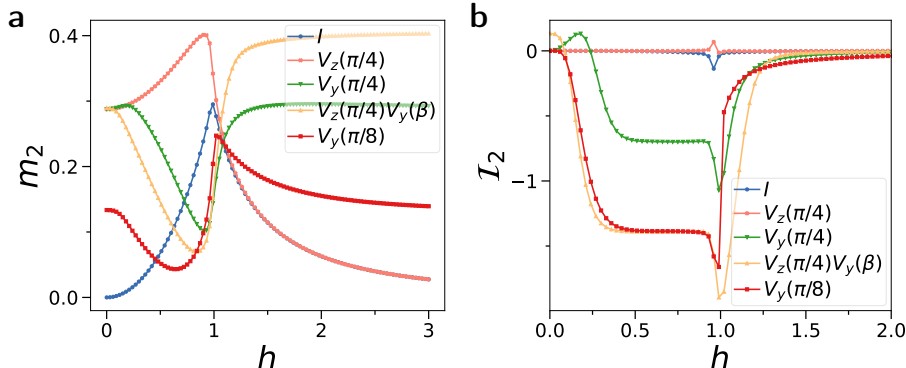

Figure 14: Magic density $m_2$ and mutual 2-SRE $\mathcal{I}_2$ close to criticality of the TFIM in various local basis $V$. Here, $\beta = \arccos(1/\sqrt{3})$ which transforms into the basis of maximal single-qubit magic. **a)** $m_2$ for various different local basis against $h$ with $N = 80$ and $\chi = 6$. **b)** $\mathcal{I}_2$ for various different local basis against $h$ with $N = 80$ and $\chi = 6$.

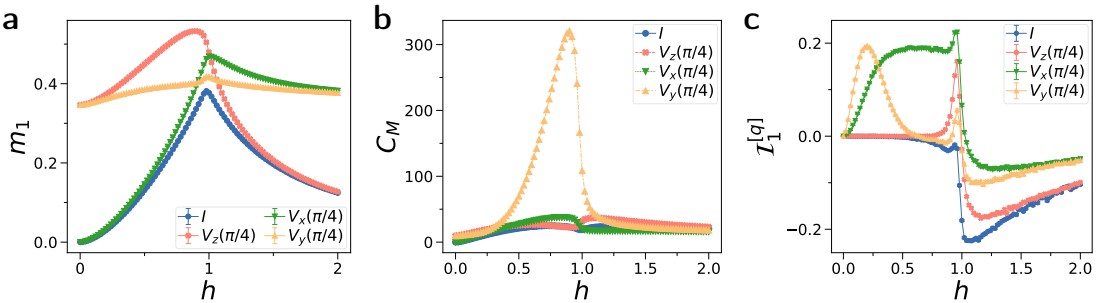

Figure 15: Magic of groundstate of TFIM against field $h$ for different basis $V_\alpha$. **a)** Von-Neumann magic density $m_1$ **b)** Magic capacity $C_M$ **c)** Mutual von-Neumann SRE $\mathcal{I}_1^{[q]}$. We have $N = 80$ qubits and $K = 10^5$ Pauli samples.

This feature is independent of the local basis. For $m_2$ depending on the chosen local basis $V$ there is no clear peak towards $h = 1$ as seen in Fig. 14a (for detailed study see Ref. [16]). In contrast, $\mathcal{I}_2$ has a clear extremum that converges towards $h = 1$ for large $N$ as seen in Fig. 14b.

Finally, we note that the mutual von-Neumann SRE $\mathcal{I}_1^{[q]}$ can also be used as basis-independent indicator of the critical point. In Fig. 15, we show von-Neumann SRE density $m_1$ (Fig. 15a), the magic capacity $C_M$ (Fig. 15b) and $\mathcal{I}_1^{[q]}$ (Fig. 15c) for different rotated basis. In Fig. 15c, We observe a distinct sharp peak for $\mathcal{I}_1^{[q]}$ close to the critical point $h = 1$ for all chosen basis, which can serve to characterize the critical point by a scaling analysis.

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
