# Peer review of "Efficient mutual magic and magic capacity with matrix product states"

_SciPost Physics, doi:SciPost Phys. 19, 085 (2025)_

## Round 2 · Referee Report · Anonymous (Referee 1) · 2025-7-5

Report
1. Parallel definitions of the distribution functions and entropies: In Sec III, their Eq.6-8 are alternative definitions of the SRE based on an alternative distribution function q=tr(rho P rho P)/2^N, in comparison with p=tr(rho P)^2/tr(rho^2)/2^N in Eq.3-4 of Sec II. They should both be placed in Sec II, before introducing the finer quantities of mutual magic and magic capacity. After all, the probability distribution is the basic of everything that follows.
2. Motivation for the alternative distribution: They should explain earlier the motivation why they need to introduce a different distribution q, otherwise it is very confusing – in principle one can define a lot of different distribution functions from a quantum state. I didnot understand their motivation until reading their paragraph following Eq.17-18 in Sec IV – is it simply because such definition saves the computation of mutual magic by roughly a factor 3 (since one does not need to evaluate A and B separately)?
3. I’m a bit confused what purpose Fig.1 serves? It is embedded in Sec IV explaining methods. But Fig.1a (with its caption and texts in paragraphs) does not tell us the efficiency and accuracy neither the physical implication of the anisotropic Heisenberg ground state. Fig.1b and its texts only tell us the convergence of small bond dimension MPS methods, and Monte Carlo based on finite bond dimension as well.
4. Sec. V-A, Clifford + T circuit physics: in their Fig.2b, the data is noisy, and there is too limited number of data points near the suspicious “crossing point”, which is far from convincing that the observable scales large for z<zc and scales to 0 for z>zc. Similarly for the data at z>zc’ in Fig.2c. It is hard to claim “critical point(s)” based on the existing data presentation in Fig.2bc. For Fig.2a, what “(entanglement/magic/information) order parameter” can distinguish the two phases if this is a “transition”?
5. Sec. V-B, in Fig.3 anisotropic Heisenberg chain ground state: this is a paradigmatic integrable model with very rich analytic and numerical understanding in the literature, concerning its universal features in long-wave-length limit. The magic calculation is new – can the authors elucidate what known or unknown universal scaling exponents can be extracted from the magic quantities they calculate? If they go to stronger anisotropic coupling to gap out the gapless quantum chain, what do they observe in the magic quantities? Can they tell the phase transition?
6. Fig.4-5 for TFIM – can they show the data collapse? Without the prior knowledge of the exact solution hc=1, what numerical hc can they get? Which scaling operators are responsible for the scaling of 1-SRE and 2-SRE here?
Recommendation
Ask for major revision
Dear Referee 1,
We thank you for your careful evaluation of the manuscript. We have attached a detailed point-by-point reply to your report. For convenience, we also include the new draft with highlighted changes.
Yours sincerely,
Poetri Sonya Tarabunga and Tobias Haug

Author: Poetri Tarabunga on 2025-08-15 [id 5735]
(in reply to Report 2 on 2025-07-17)We thank the Referee for their very positive assessment of our work. Their recommendation for publication is highly appreciated. We have added all the suggested papers in our citations.

---

## Round 2 · Referee Report · Anonymous (Referee 2) · 2025-7-17

Strengths
- Well written and accessible
- Introduction of well-motivated quantities in magic resources for many-body models
Weaknesses
Report
The numerical results are clear and relevant: mutual SREs behave as robust probes of criticality (also under basis rotation), and the magic capacity connects naturally to the anti-flatness of the Pauli spectrum, offering an operational handle on sampling complexity.
The paper is technically solid and well written. The methods build on prior developments—especially those on stabilizer Rényi entropies and Pauli sampling—but refine them meaningfully and apply them to both ground-state transitions and random circuit ensembles. The Monte Carlo improvements and hybrid sampling approaches are nicely done and practically useful.
Requested changes
To better situate the work in the current literature, the authors should cite: - arXiv: 2501.18679 - arXiv: 2503.07468 - arXiv:2412.10229 - arXiv:2408.16047 - arXiv:2502.20455 - arXiv: 2305.11797
These papers are relevant for the discussion of Pauli statistics, randomness in Clifford+T circuits, and resource theory of magic in many-body systems.
Recommendation
Publish (easily meets expectations and criteria for this Journal; among top 50%)

---

## Round 3 · Referee Report · Anonymous (Referee 1) · 2025-8-29

Report

I am satisfied by the authors' response and revision of the manuscript. I recommend this manuscript to be published.

Recommendation

Publish (meets expectations and criteria for this Journal)

---

## Round 3 · Author Response

Dear Editor,

Thank you for managing our submission, and for informing us about the reports provided by the Referees. We appreciated the Referees' comments on our work. We have improved the manuscript according to their comments. We hope that our revised manuscript is suitable for publication in SciPost Physics.

Yours sincerely,
Poetri Sonya Tarabunga and Tobias Haug

---

## Round 3 · List of Changes

• We reorganized Sec. II and III. The alternative distribution $q_\rho$ and its corresponding SRE
is now introduced in Sec. II.
• Added Fig.S5 and Fig.S6d to demonstrate determining the critical field of the transverse-
field Ising model
• Added Appendix G to explain fitting procedure for determining critical field
• Expanded Fig.2 into Fig.2 and Fig.3, with more samples to reduce noise, as well as in-
depth study of universal behavior of the C_M and m_1.
• Added new Appendix A to provide proof of marginals for probability distribution $q_\rho$

---

## Editorial Decision

published